# Birth season shapes the infant metabolome and development in Tanzania: a secondary explorative analysis of the early life interventions for childhood growth and development in Tanzania (ELICIT) trial

Elizabeth A. Wimborne[1], Daniela Hampel[2,3], Lindsay Allen [3], Daan R. van der Veen [4], Gordana Panic [1,5,6], Madhuri Panchal[1], James A. Platts-Mills[7], Estomih Mduma[8], Ladislaus Blacy[8], Samwel Jatsoh[8], Paschal Mdoe[8], Rebecca Scharf[9], Mark D. DeBoer[9] & Jonathan R. Swann [1] ✉

Rural communities in resource-limited settings are exposed to seasonal patterns in food insecurity. The pre- and early post-natal period is a key developmental window, sensitive to nutritional availability and quality. While season of birth has been associated with differences in epigenetic, developmental and health outcomes, it is unclear if this imprints on the developing metabolome and contributes to adverse phenotypic outcomes. Here we show that urinary and plasma metabolites in Tanzanian infants follow seasonal waveforms dependent upon their month of birth, persisting up to at least 18 months of life, which were related to food insecurity, breastmilk composition and rainfall. This includes developmentally relevant metabolites, including choline, trimethylamine-*N*-oxide, and polyunsaturated fatty acids. Cognitive measures at 18 months also followed seasonal waveforms based upon month of birth which correlated with seasonal metabolites. Additionally, variation in metabolic status modulated the effectiveness of a maternal-infant nicotinamide intervention targeting stunting. Our findings show that seasonal environmental pressures shape mother-infant dyad food insecurity with lasting ramifications for the infant metabolome and development. These findings support the need for season-dependent nutritional and lifestyle interventions targeting developmental shortfalls in these communities.

The pre- and post-natal early life environment imprints on the infant with enduring ramifications for health and development[1]. Nutritional availability and quality, as well as enteric infections, are major contributors, and in many settings follow seasonal variation. Emerging evidence shows seasonal patterns in food insecurity during the pre- and post-natal periods influence the developing epigenome of the infant with implications for later life phenotypes, including cardiometabolic health, mortality, and body composition[2-4]. However, it is currently unclear if these seasonal patterns impart lasting effects on the infant metabolome and on development. Previous metabolomic

studies highlight a diverse array of metabolic derangements associated with early life acute and chronic undernutrition and their related outcomes such as stunting, wasting, and cognitive impairments[5–8]. This includes disruptions to amino acid, choline, tryptophan-nicotinamide, and gut microbial metabolism. Given the limited effectiveness of nutritional and lifestyle interventions to attenuate developmental shortfalls[9–12], understanding seasonal pressures on the developing metabolic phenotype may explain differences in treatment responses. Climate change has been shown to exacerbate seasonal variations in environmental conditions and is anticipated to increase the frequency and severity of droughts and food insecurity[13]. As such, understanding the detrimental impact of adverse climatic conditions on early life undernutrition and its impact on development remains crucial.

To investigate this, we studied infants over the first 18 months of life in Haydom, a rural agricultural town in Tanzania. Here, communities experience seasonal patterns in food insecurity due to their unimodal crop cycle. In this setting, food insecurity decreases following the harvest of beans (March) and maize (June)[14,15] but rises during the pre-harvest period (November-February) when harvest stocks become depleted, and diet diversity reduces. Enteric infections are common in this population, and certain pathogens, such as Shigella, vary seasonally, having the greatest burden in the rainy season. These infants were enrolled in the Early Life Interventions for Childhood growth and development in Tanzania (ELICIT) randomized control trial[16–18]. This trial had a 2 × 2 factorial design including nutritional nicotinamide and antimicrobial interventions, provided individually and in combination. However, no effect on the primary outcome of 18-month length was observed. Here, we studied infants from the placebo group and explored whether the month of birth imprints on the urinary and plasma metabolomes at six, 12, and 18 months of life and whether metabolic differences related to variation in developmental outcomes, namely growth and cognition. Next, we assessed if seasonal patterns in food insecurity and maternal breastmilk components (i.e., vitamins, human milk oligosaccharides (HMOs), metabolites, and lipids) contributed to these patterns.

In areas with maize-based diets, deficiency of the essential amino acid tryptophan is common and has been associated with infant stunting[19]. Tryptophan is key for development, being necessary for protein, muscle, and neurotransmitter production. Additionally, through its metabolism via the kynurenine pathway, it is important for the synthesis of nicotinamide adenine dinucleotide (NAD+), an essential co-enzyme for energy production and growth. We, and others, have shown that the tryptophan-niacin-NAD+ pathway is related to growth and perturbed with early life undernutrition and infections[5,11]. This is due to reduced precursor availability (niacin, nicotinamide, tryptophan, nicotinamide riboside) and immune-associated disruptions (induction of indoleamine 2,3-dioxygenase activating the kynurenine pathway)[16,20]. To prevent deficiencies in this pathway, the ELICIT study included a group of infants who received nicotinamide as a nutritional intervention (nicotinamide group). This was provided via breastfeeding mothers for the first six months of life before direct supplementation to the infant until 18 months of life. As no improvement in length-for-age z-score at 18 months was observed, we investigated whether seasonal fluctuations in the metabolomic profiles across infants explained variation in responsiveness to nicotinamide.

In this work, we show that infant urinary and plasma metabolites in Tanzanian infants follow seasonal waveforms dependent upon their month of birth, persisting up to at least 18 months of life. Such metabolic variation reflected seasonal changes in food insecurity, breastmilk composition and rainfall. Cognitive measures at 18 months also displayed seasonal trends based upon birth month, which correlated with seasonal metabolites. Additionally, variation in metabolic status modulated the effectiveness of a maternal-infant nicotinamide intervention targeting stunting. Understanding circannual factors shaping the developing infant system may assist in tailoring interventions to increase their effectiveness against adverse outcomes associated with childhood undernutrition.

## Results

### Participants

This is a secondary analysis of the ELICIT cohort[16], following infants born between September 2017–18 longitudinally to 18 months of life. The ELICIT study was a randomized 2 × 2 factorial, double-blind, placebo-controlled trial with the aim of improving early life growth using a nicotinamide and/or antimicrobial (azithromycin, nitazoxanide) intervention. Nicotinamide was provided to breastfeeding mothers (250 mg daily tablets for six months post-delivery), followed by direct infant supplementation (100 mg sachets) from six to 18 months.

Here, data from the placebo group ($n = 278$) and infants receiving nicotinamide ($n = 276$, breastfeeding mothers provided 250 mg daily tablets for six months post-delivery, followed by direct infant supplementation (100 mg sachets) from six to 18 months) was studied. Urine was sampled at six months, and urine and blood were sampled at 12 and 18 months. A schematic, study design overview and analysis strategy are provided in Figs. S1, S2, and Supplementary Materials (Statistical Analysis Plan), respectively. This study included infants of various nutritional states, including stunted (length-for-age Z-score [LAZ] < −2), underweight (weight-for-age Z-score [WAZ] < −2), and normal growing infants (Supplementary Data. 1). The antimicrobial group was not considered for this analysis, as the biological samples were collected prior to the administration of a single-dose of the antimicrobials at six, 12, and 18 months and their long-term biochemical consequences cannot be excluded. Additionally, receipt of non-study antibiotics was common (54 − 86% infants).

### Birth month influences infant metabolome

To determine whether the infant metabolome exhibits patterns dependent upon month of birth, targeted liquid chromatography-mass spectrometry (LCMS)-based metabolomics was performed on plasma samples from infants at 12 months of life from the placebo arm of the ELICIT study ($n = 199$). Samples were collected from infants born throughout the year (collected from September 2018 to September 2019), and a total of 628 plasma metabolites were measured. Cosinor analysis[21] was used to identify molecules whose concentrations followed a seasonal waveform across the population based upon the infant's month of birth (Fig. 1A–C). A cosine wave with a sloping mesor was fitted to the data using non-linear curve fitting solved by least squares, and statistically compared to H0 of solely a sloping line model using an extra-sums-of-squares $F$-test. A total of 46 circulating metabolites exhibited birth month dependent variation in their concentrations at 12 months (Supplementary Data. 2). This included choline, trimethylamine-$N$-oxide (TMAO), eicosapentaenoic acid (EPA), eicosadienoic acid, 17 phosphatidylcholines, six ceramides, three amino acids (aspartate, glutamine, glutamate), 5-aminovaleric acid (5-AVA), γ-aminobutyric acid (GABA), methionine sulfoxide, two cholesterol esters, 3-hydroxyglutaric acid, proline betaine, sphingomyelin C26.1, three hexosylceramides, three triglycerides, and octadecenoylcarnitine. Of these, seven remained significant (choline, TMAO, EPA, methionine sulfate, cer(d18:2/14:0), TG 20:0_34:1 and hex2cer(d18:1/18:0); Figs. 1B, S3) following adjustment for covariates including enrollment WAZ, socioeconomic status, number of months of exclusive breastfeeding, and maternal factors (weight, height, age) and following correction for multiple testing ($q < 0.15$; $n = 3$ for $q < 0.10$). The birth month of infants with the greatest concentration of each metabolite is determined by the peak phase of the wave. These are provided for each metabolite in Supplementary Data. 3. For example, at 12 months, February-born infants had the highest circulating aspartate, while those born six months later in August had the lowest.

 

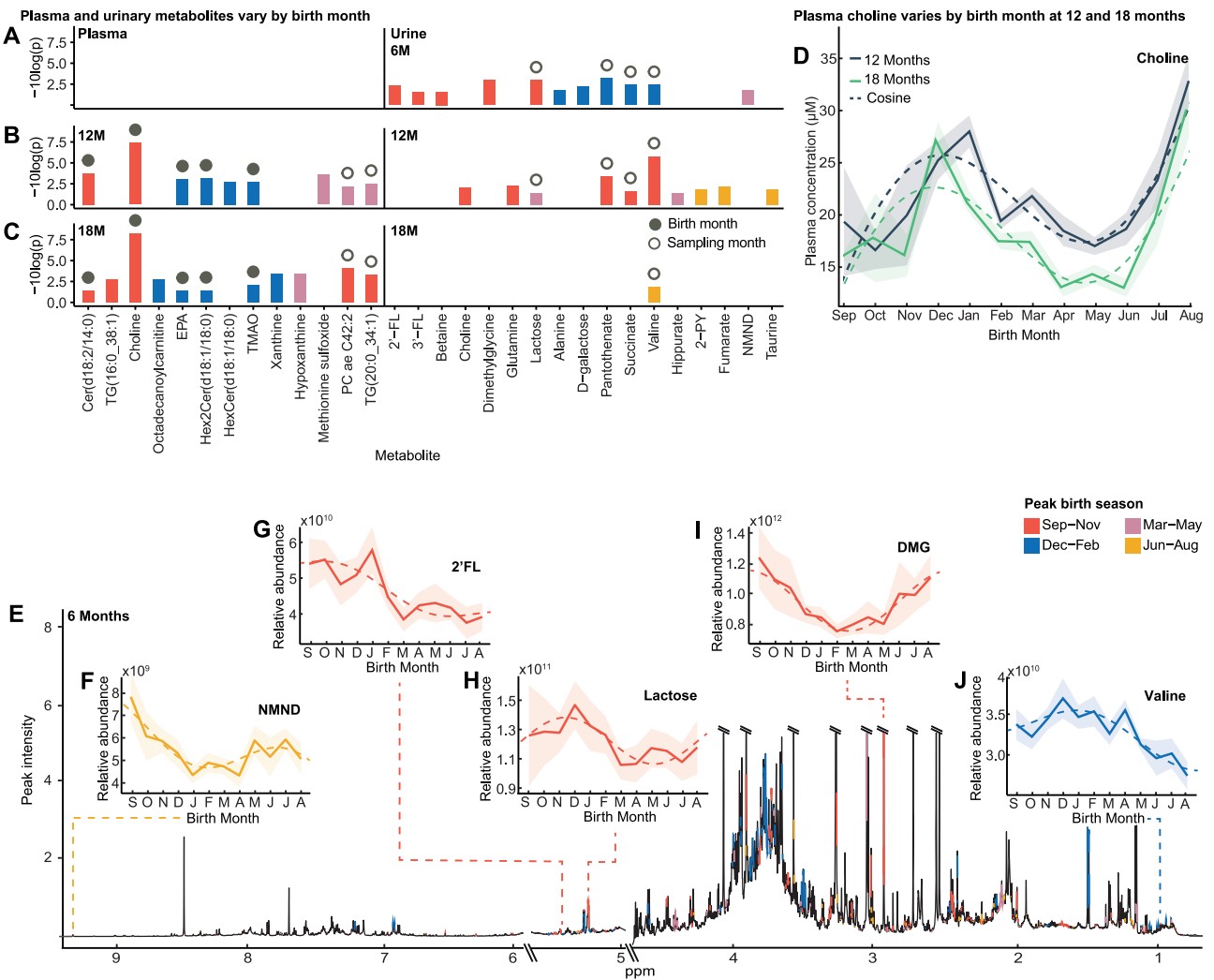

**Fig. 1 | Systemic and urinary metabolome variation at 6, 12 and 18 months is influenced by the month of birth. A−C** Bar chart of 6-, 12- and 18-month urinary, and 12- and 18-month plasma metabolites that follow a seasonal waveform based upon month of birth, from cosinor analysis (plasma $n = 199$; urine 6-month $n = 278$; 12-month $n = 270$; 18-month $n = 266$; $p < 0.05$ & $q < 0.15$ following Benjamini-Hochberg correction at ≥ one sampling point by extra-sums-of-squares $F$-test; exact $p$-values are provided in Supplementary Data. 2, 3). Color indicates birth season of peak abundance (September-November, red; December-February, blue; March-May, pink; June-August, yellow). Solid circle represents birth-month-dependent metabolites (±1.5-month difference between birth month of peak abundance at two sampling points), unfilled circle indicates sampling-month-dependent (>four-month difference). **D** Mean plasma choline concentration by birth month (bold line; $n = 199$) at 12 (gray) and 18 months (green) following cosine distribution (dashed line) with 95% CI overlaid (shaded). **E** Median [1]H NMR six-month urinary metabolic spectral profile ($n = 278$). Metabolites identified to significantly fit cosine waveforms based upon birth month are colored by birth season of peak abundance. Urinary metabolite relative abundance obtained by integrating the area under spectral regions. **F−J** Examples of urinary metabolites at six months that follow a cosine distribution (dashed line) by birth month. Mean relative abundance with 95% CI overlaid (shaded). Source data are provided as a Source Data file. EPA, eicosapentaenoic acid; TMAO, trimethylamine $N$-oxide; 2-PY, $N$-methyl-2-pyridone-5-carboxamide; NMND, $N$-methyl nicotinamide; 2′-FL, 2′-fucosyllactose; 3′FL, 3-fucosyllactose; DMG, dimethylglycine.

To determine if these metabolic cycles were driven by contemporary pressures at 12 months of life (i.e., time of sampling) or a legacy of the month of birth, plasma was sampled six months later at 18 months. Several plasma metabolites exhibited circannual patterns across the infants at 18 months ($n = 27$; $n = 6$ following FDR correction ($q < 0.15$; $n = 5$ for $q < 0.10$) and adjustment for confounders; Fig. 1C, Fig. S3 and Supplementary Data. 3). Metabolites whose peak phase remained static between sampling points (±1.5-months to account for season) were considered to be influenced by time of birth (Fig. S1). For instance, choline is birth-month-related as November-December born infants persistently had the highest circulating choline at 12 and 18 months, while those born May-June had the lowest (Fig. 1C). These patterns were maintained despite variable pressures across these sampling points, including food insecurity and rainfall. Conversely, metabolites influenced by sampling time were identified as those with

a ~six-month (four-eight months) phase difference between the 12- and 18-month samples and so exhibit seasonal variation within individuals (Fig. S1). Interestingly, 12 plasma metabolites exhibited seasonal patterns at both sampling points including choline, TMAO, EPA, glutamate, eicosadienoic acid, cer(d18:2/14:0), hex2cer(d18:1/18:0), TG(20:0_34:1), octadecenoylcarnitine, methionine sulfoxide, and three phosphatidylcholines (PC ae C42:2, PC ae C42:3, PC ae C40:2). Of these, seven remained significant following FDR correction in at least one time point. Here, there were five birth-month-related metabolites including choline, TMAO, EPA, hex2cer(d18:1/18:0), and cer(d18:2/14:0), and two sampling-month-dependent metabolites, TG(20:0_34:1) and PC ae C42:2 (Fig. S3). After adjustment for covariates (enrollment WAZ, socioeconomic status, number of months of exclusive breast-feeding, and maternal factors (weight, height, and age), the variation in EPA and hex2cerd(18:1/18:0) was no longer significant at 18 months.

Additionally, untargeted [1]H-nuclear magnetic resonance (NMR) spectroscopy-based metabolomics was performed on urine samples collected from the placebo group infants at six, 12, and 18 months of life (six-month, $n = 278$; 12-months, $n = 270$; 18-months, $n = 266$; Fig. 1A–C). At six months (Fig. 1E), 11 urinary metabolites were identified to exhibit variation across the infants based upon the month in which they were born. These included N-methyl nicotinamide (NMND; Fig. 1F), pantothenate (vitamin B5), breastmilk-related metabolites (HMOs (2′-fucosyllactose (2′-FL; Fig. 1G), 3′-fucosyllactose (3′-FL)), lactose (Fig. 1H), and D-galactose), betaine, dimethylglycine (DMG; Fig. 1I), valine (Fig. 1J), alanine, and succinate. Of these nine remained significant when and adjusted for the covariates enrollment WAZ, number of months of exclusive breastfeeding, maternal age and socioeconomic status (pantothenate, dimethylglycine, lactose, succinate, valine, 2′-FL, NMND, D-galactose, and 3′FL) and corrected for multiple testing ($q < 0.15$; $n = 4$ for $q < 0.10$; Figs. 1A, S4 and Supplementary Data. 3). Interestingly the seasonality was dampened in the HMO 2′-FL when adjusted for maternal body composition (weight, height). Additionally, at 12 months, the excretion of four metabolites significantly varied by birth month ($q < 0.15$; $n = 4$ for $q < 0.10$; Fig. 1B, Supplementary Data 3). The peak phases of three metabolites, valine, pantothenate and succinate, shifted by ~six months between sampling points and so were considered sampling-month-dependent.

A PCA model was constructed with the seasonal metabolites to compare infants born during different three-month windows (e.g., Sep-Nov, Dec-Feb, Mar-May, Jun-Aug) at six, 12 and 18 months. Permutational multivariate analysis of variance (PERMANOVA) confirmed significant metabolic differences between the windows (adonis $p = 0.001–0.04$, Fig. S5). Similarly, significant pairwise PLS-DA models were obtained comparing seasonal metabolites across infants born in opposing time windows (AUC 0.67–0.88; Fig. S5; metabolite VIP scores listed in Supplementary Data. 4). For metabolites identified as exhibiting seasonal variation across the population, the mean concentrations and abundances by birth season are provided in Supplementary Data. 5.

After stratifying by sex, no plasma or urine metabolites exhibited circannual patterns in the females, while two, 17, and six metabolites were noted in the males at six, 12 and 18 months respectively after FDR and covariate adjustment (enrollment WAZ, socioeconomic status, number of months of exclusive breastfeeding, and maternal factors (weight, height, age)) (Fig. S6, Supplementary Data. 6; for $q < 0.10$, 6 M $n = 1$; 12 M $n = 11$; 18 M $n = 5$). Here, plasma choline and TMAO and urinary D-galactose were noted to be birth-month-dependent metabolites in males, while urinary pantothenate and lactose were sampling-month dependent.

## Metabolomic seasonality on cognition

To determine the developmental significance of this biochemical seasonality, we explored relationships with cognitive outcomes. Child development was evaluated at 18 months using the Malawi Development Assessment Tool (MDAT)[22]. This assessed gross and fine motor, language, and social skills. Cosinor analysis identified that language scores ($R^2 = 0.24$) and total MDAT ($R^2 = 0.13$) measured at 18 months varied across infants by birth month, where those born in January and February respectively had the greatest scores (Fig. 2A). When stratified by 'Water and sanitation, Assets, Maternal education and household Income' (WAMI) scores[23], a measure of socioeconomic status, this pattern persisted only in those with the top 50% scores ($R^2 = 0.22$). When stratified by sex, this variation was only observed in the female population ($R^2 = 0.31$). Cross-correlation analysis identified seven seasonal 12-month plasma metabolites that significantly correlated with language scores six months later (Fig. 2B, Supplementary Data. 7). Those infants with the greatest language scores had the highest circulating TMAO (Fig. 2C), EPA (Fig. 2D), PC ae C42:2, TG(20:0_34:1),

methionine sulfoxide, hex2cer(d18:1/18:0) and hexcer(d18:1/18:0) at 12 months.

## Environment shapes seasonal metabolites

Next, we explored whether fluctuating nutritional availability during early life contributed to this metabolic seasonality. Birth anthropometry reflects environmental and nutritional pressures faced by the mother-fetus prior to and during pregnancy and relates to long-term growth outcomes[24]. We have previously shown that infants born in the five-month preharvest period (November-March 2017–2018) had lower enrollment weights than those born in the seven months postharvest (April to October)[18]. However, enrollment WAZ and LAZ did not follow 12-month seasonal waveforms, even after stratifying by WAMI.

As rainfall contributes to harvest yield, seasonality analysis was applied to Haydom rainfall data from the prenatal year (2016–2017), birth year (2017–2018) and to the 8-year average (2010–2018) (Fig. 3A). All measures significantly fitted a seasonal waveform with peak phases (highest rainfall) in January-February, reflecting annual climatic patterns. Similarly, monthly maternal food insecurity reports across an 18-month period from birth (reported as worry; expressed as % mothers) also followed a seasonal waveform peaking in January (Fig. 3B). These food insecurity patterns occur as harvest stocks deplete and food prices rise during the rainy season (November-May)[14,25]. Conversely, the dry season (June-October) associates with reduced food insecurity following the harvest of beans in March and maize in May (Fig. 3B). Interestingly, rainfall in the previous year cross-correlated with reported food insecurity (S.Text.1).

To explore the influence of maternal food insecurity at birth on these seasonal metabolites sampled at 6–18 months of life, cross-correlation analysis was performed. Phase lag represents the time difference between the month of peak food insecurity and the birth month of peak metabolite abundance (Fig. S1). Four birth-month-related metabolites (consistent pattern between sampling times) significantly correlated with food insecurity, including choline, TMAO, hex2Cer(d18:1/18:0), and the omega-3 PUFA EPA. Minimal phase lags (0–40 days) were observed between their peak abundances and peak food insecurity at birth, and this pattern persisted at 12 and 18 months for choline, hex2Cer(d18:1/18:0), and EPA (Fig. 3D, E, Supplementary Data. 8). These were all highest in infants born during peak food insecurity (November-February). Additionally, food insecurity at birth cross-correlated with several seasonal metabolites at six and 12 months with a range of phase lags (Fig. 3C, D, Supplementary Data. 8 and S.Text.2). Several metabolites displayed the highest abundance in infants born during periods of minimal food insecurity, with a ~six-month phase lag. At six months, this included the urinary excretion of NMND, a marker of nicotinamide availability used in NAD[+] synthesis; betaine and DMG. Several metabolites also aligned with contemporary food insecurity at the time of sampling (Fig. S7A–C, Supplementary Data. 9 and S.Text.3).

## Seasonal variation in breastmilk

In total, 81% of infants were exclusively breastfed up to six months of life, and seasonal food insecurity patterns acted indirectly on the infant via breastmilk throughout this period. We therefore explored breastmilk composition seasonality at the vitamin, HMO and metabolome-lipidome level. Of those exhibiting seasonality across the population, the mean metabolite concentration per season is documented in Supplementary Data. 10.

A total of 19 B vitamin-related metabolites were measured in breastmilk from 292 mothers, sampled at one- and five-months postpartum. Cosinor analysis identified four metabolites at one-month, and seven at five months that displayed seasonality based on sampling month, which persist following correction for multiple testing (Fig. 3F, Supplementary Data. 11). Of these, one metabolite exhibited

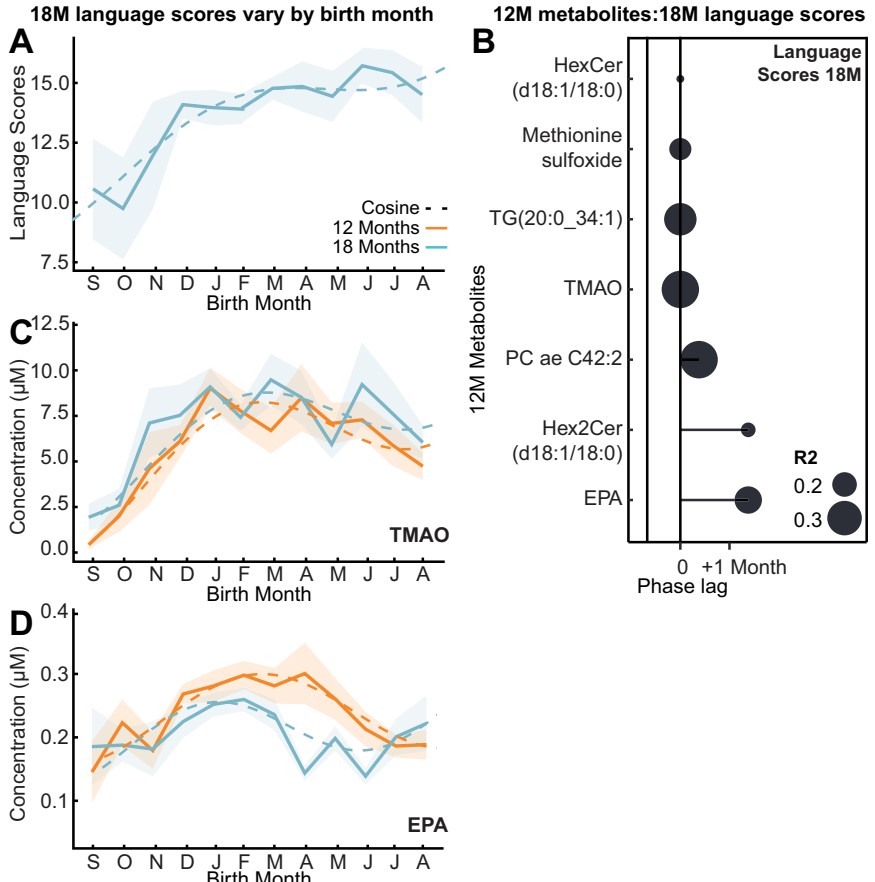

**Fig. 2 | Influence of birth seasonality on cognitive outcomes and related metabolites. A** Mean language MDAT (Malawi Developmental Assessment Tool) score by birth month at 18 months (bold line; $n = 199$) following cosine distribution (dashed line; $p = 0.0006$) with 95% CI overlaid (shaded). **B** Lollipop plot visualizing phase lags (months) between birth month of greatest 18-month language MDAT scores and greatest seasonal 12-month plasma metabolites ($n = 199$; Benjamini-Hochberg corrected $q < 0.05$; exact $p$-values are provided in Supplementary Data. 7) from cross-correlation analysis. Size indicates $R^2$. **C, D** Mean concentration of plasma trimethylamine N-oxide (TMAO) and eicosapentaenoic acid (EPA) by birth month ($n = 199$) following cosine distribution (dashed line) at 12 (orange) and 18 months (blue) with 95% CI overlaid (shaded). Source data are provided as a Source Data file.

seasonality at both sampling times, where biotin (vitamin $B_7$) was consistently highest from mothers who delivered in April (sampled May at one month and September at five months), a period of increasing food availability following the bean harvest in March. Nicotinamide riboside (NR) and nicotinic acid (NAc), precursors involved in the $NAD^+$ synthesis pathway, exhibited seasonality in breastmilk. NR at one-month, and $NAD^+$ at five months, were highest in milk from mothers delivering in April (NR sampled May, $NAD^+$ sampled September), whilst NAc was greatest in five-month breastmilk collected from mothers sampled in July.

The metabolomic-lipidomic signatures of a subset of breastmilk samples (54 individuals) collected at one- and five-months postpartum were characterized alongside their HMO profiles (19 compounds). Due to low sample size, cosinor analysis could not be implemented. Samples were therefore grouped by season of sampling (Sep-Nov, Dec-Feb, Mar-May, Jun-Aug) and variation was assessed by ANOVA or Kruskal-Wallis, and post-hoc tests (FDR adjusted). Mothers sampled between September – November were observed to have the greatest 2′-FL, lacto-N-fucopentaose I (LNFP-1) and glutamine in their milk at one-month postpartum. Additionally, 2′-FL and LNFP-1 were greatest in five-month milk sampled from mothers in June-August (Supplementary Data. 12). Interestingly, no birth-month-related metabolites (choline, TMAO, EPA, hex2-cer(d18:1/18:0), and cer(d18:2/14:0)) were observed to fluctuate across the year in the breastmilk, suggesting intake via breastmilk does not directly drive these alterations (Fig. S8).

We next assessed the impact of the nutritional environment of the mother on the breastmilk metabolome. Cross-correlation was performed between seasonal breastmilk B-vitamin-related metabolites and contemporary food insecurity. (Fig. 3F, Supplementary Data. 13). Short phase lags (10–90 days) were noted between peak food insecurity and maximal breastmilk biotin, and NR at one-month and flavin mononucleotide, at five months. Metabolites most abundant in milk sampled during periods of high food availability (110–200 days phase lag from peak food insecurity) included pyridoxamine at one-month; pantothenic acid (vitamin $B_5$), Nac, NAD, pyridoxal-5-phosphate, and biotin, at five months.

To assess the impact of food insecurity on the breastmilk metabolome and HMOs measured from the subset of mothers, we explored biochemical variation in the season-dependent metabolites associated with contemporary maternal food insecurity reports (No food insecurity [$n = 40$], reported food insecurity [$n = 14$]; Wilcoxon test $q < 0.05$; Fig. 3G, Supplementary Data. 14). At one-month, 2′-FL and LNFP-1 were lower in milk from mothers who did not report food insecurity. At five months, only three mothers from this subset reported food insecurity; therefore, statistical analysis was not performed. Additionally, to explore transmission of these seasonal patterns to the infant, cross-correlation analysis compared five-month breastmilk vitamins and six-month urinary metabolites. Significant correlations were observed with a range of phase lags (Fig. S9, Supplementary Data. 15). These included no phase lag between pantothenic acid in the five-month breastmilk and the six-month urine,

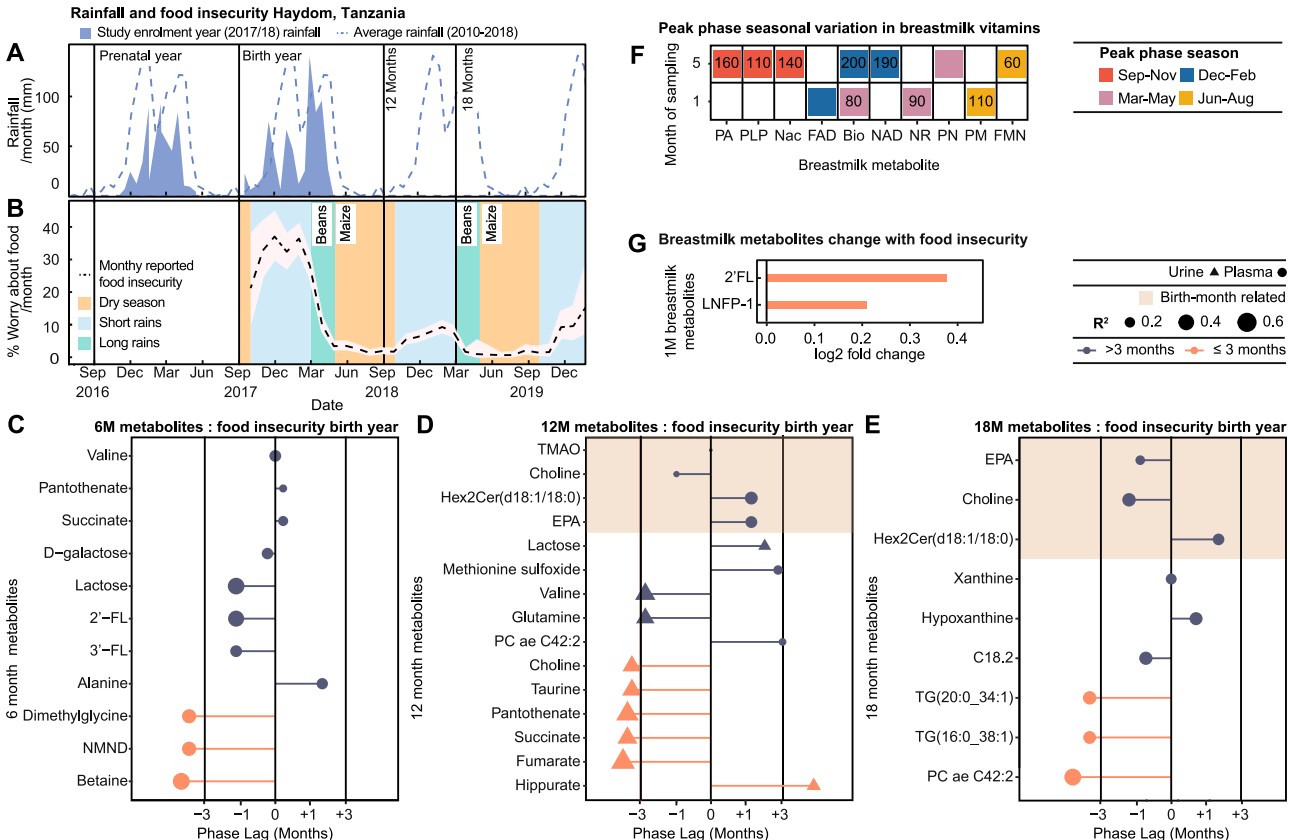

**Fig. 3 | Annual variation in environmental factors across pre- and postnatal periods contributes to metabolic seasonality. A** Mean monthly (shaded) and 8-year average rainfall (dashed line) in Haydom, Tanzania. **B** Mean monthly food insecurity (% mothers worried about food availability; black line; 95% CI shaded) reported over an 18-month period following enrollment. Annual patterns of rain and harvests highlighted (dry season, yellow; short rains, blue; long rains, green). **C–E** Lollipop plots displaying phase lags between peak food insecurity during birth year (January 2018) and the birth month of greatest plasma and urinary metabolite abundance at 6 ($n = 278$), 12 ($n = 270$) and 18 months ($n = 266$; $q < 0.05$) from cross-correlation. Color indicates phase lag (>3 months, purple; ≤3 months, pink); shape indicates source (urine, triangle; plasma, circle), and size $R^2$. Shaded area highlights birth-month-dependent metabolites. **F** Heatmap of 1- and 5-month breastmilk metabolites displaying seasonality based on sampling month from cosinor analysis

($n = 292$; $q < 0.15$). Colors indicate highest abundance sampling season (September-November, red; December-February, blue; March-May, pink; June-August, yellow). Values indicate phase lags (days) between peak food insecurity and peak metabolite abundance from cross-correlation ($q < 0.05$). **G** Bar chart visualizing log2 fold change of 1-month breastmilk metabolite expression by reported food insecurity at sampling (No $n = 40$; yes $n = 14$). Differences assessed by two-sided Wilcoxon test with Benjamini Hochberg correction for multiple testing ($q < 0.05$; 2-fucosyllactose (2'FL) $p = 0.0004$; lacto-*N*-fucopentaose I (LNFP-1) $p = 0.0006$). Source data are provided as a Source Data file. FAD flavin adenine dinucleotide, FMN flavin mononucleotide, PLP pyridoxal phosphate, Nac nicotinic acid, Pa pantothenic acid, NAD nicotinamide adenine dinucleotide, NR nicotinamide riboside, PM pyridoxamine, PN pyridoxine, GABA γ-aminobutyric acid, 3'FL 3-fucosyllactose, NMND *N*-methylnicotinamide, EPA eicosapentaenoic acid, C18.2, octadecadienylcarnitine.

with the greatest abundance of this vitamin noted in February/March-born infants. This demonstrates the imprinting of seasonal patterns in breastmilk composition on the metabolic signatures of the infant.

Breastfeeding has been associated with increased cognitive development[26], therefore, relationships between season-dependent breastmilk metabolites and cognitive outcomes were explored (Spearman's rank correlations; $q < 0.05$; Fig. S10, Supplementary Data. 16). Here, 18-month language scores were negatively associated with two seasonal breastmilk metabolites, 2'-FL and LNFP-1, at one and five months.

**Metabolomic seasonal variation on growth**

As with enrollment anthropometry, no significant seasonal waveforms were detected across anthropometric measures (LAZ, WAZ, HCZ [head circumference-for-age Z-scores], and MAZ [mid-upper arm-for-age circumference (MUAC) Z-scores]) by birth month from three-18 months (Supplementary Data. 17). We next assessed relationships between seasonal metabolites and growth measures (WAZ and LAZ) (Fig. S11, Supplementary Data. 18). Correlation analysis found a trend for a positive linear relationship between six-month urinary NMND and six-month LAZ ($p = 0.007$; $q = 0.08$). This was

confirmed using time series analysis, where infants persistently in the top tertile for urinary NMND excretion at six-, 12-, and 18-months experienced a less pronounced drop in LAZ growth trajectory from zero-18 months of life ($p = 0.001$, 95% CI: $p = 0.0002$, 0.006; Fig. 4A).

**Supplementing infants with nicotinamide**

Given the positive associations between urinary NMND and postnatal growth[5], the impact of nicotinamide supplementation on infant anthropometry was assessed in this setting as part of the ELICIT RCT (nicotinamide group)[16,17]. Mothers ($n = 298$) received 250 mg daily nicotinamide tablets for six months post-delivery while breastfeeding, followed by direct infant supplementation (100 mg sachets) from six to 18 months. Maternal nicotinamide supplementation significantly increased breastmilk NR and ablated the seasonal variation of this metabolite (Figs. 4B, C, S12A-B). Additionally, six-month NMND excretion by the infants of supplemented mothers was increased, and the seasonality was lost (Figs. 4D, E, S12C). Despite these metabolic changes, nicotinamide supplementation did not alter anthropometry between the supplemented infants and those not receiving nicotinamide[17].

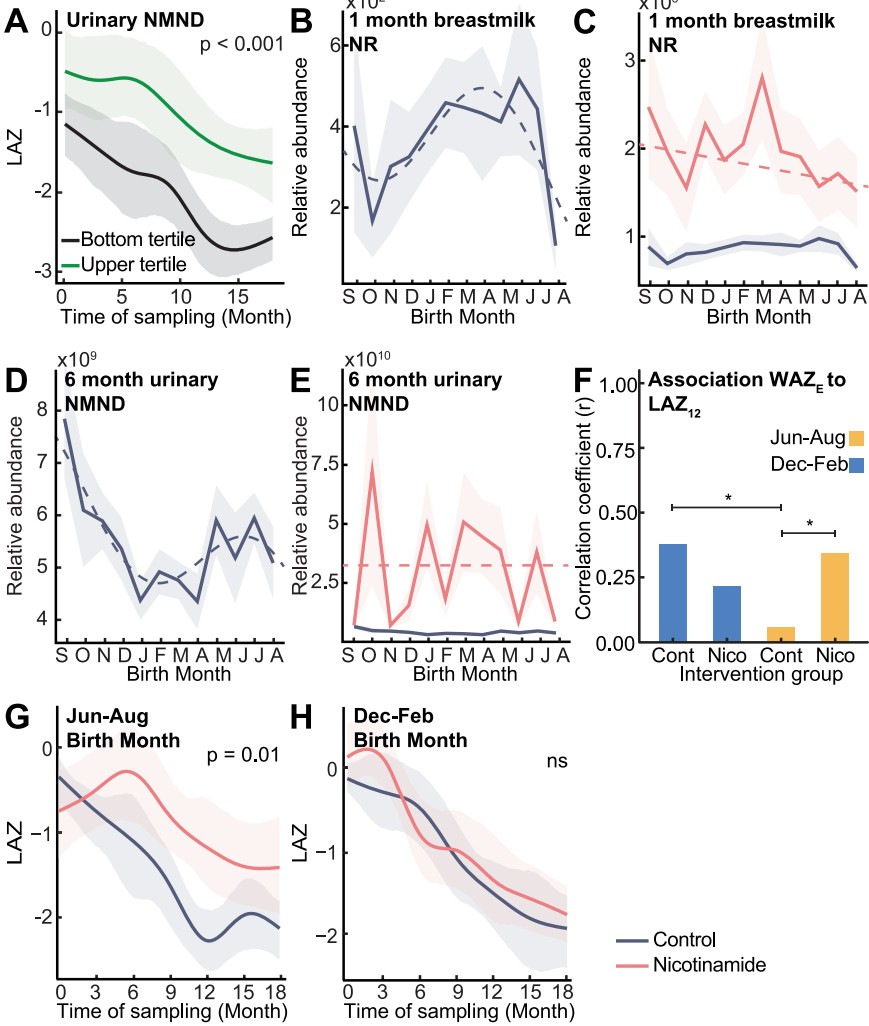

**Fig. 4 | Nicotinamide availability influences infant growth and can be improved through maternal and infant supplementation. A** Differences in 0–18-month LAZ (length-for-age Z-score) of non-supplemented infants persistently in upper ($n = 22$; green) and lower ($n = 16$; black) urinary N-methyl nicotinamide (NMND) tertiles (6, 12 and 18 months) assessed by time series analysis. Mean relative abundance of (**B**, **C**) 1-month breastmilk nicotinamide riboside (NR) and (**D**, **E**) 6-month urinary NMND following cosine (dashed line) with 95% CI (shaded) in control infants (NR $n = 292$, NMND $n = 278$; purple) and nicotinamide supplemented (NR $n = 295$, NMND $n = 276$; pink), colored by intervention group. **F** Correlation coefficient between enrollment WAZ (weight-for-age Z-score; WAZ$_E$) and 12-month LAZ (LAZ$_{12}$) for control and nicotinamide-supplemented infants, colored by birth season (December-February, blue; June-August, yellow).

Significance from Fisher's Z-transformation comparing correlation coefficients (two-sided Z-test; December–February control vs June–August control, $p = 0.025$; June–August control vs nicotinamide, $p = 0.044$; * indicates $p < 0.05$; Nicotinamide December–February born $n = 114$, control $n = 116$; nicotinamide June–August born $n = 55$, control $n = 55$). **G**, **H** LAZ trajectories from 0–18 months for children in the top enrollment WAZ tertile (WAZ ≥ −0.13) born in June–August (nicotinamide $n = 21$, control $n = 18$) and December–February (nicotinamide $n = 30$, control $n = 21$) assessed by time series analysis. Group mean trajectories (solid line; colored by group) with 95% CI (shaded area); $p$-values from permutation test on the area between group mean curves (1000 permutations; June–August $p = 0.013$). Source data are provided as a Source Data file.

We next explored the influence of seasonality on the effectiveness of supplementation. December-February-born infants experienced low food insecurity from three to nine months old. These infants show a strong relationship between enrollment WAZ and 12-month LAZ ($n = 116$; $r = 0.38$, $q < 0.001$; Figs. 4F, S13), indicating that when food is not limiting, the starting weight of the infant influences future growth. This observation was not significant when infants from the lower 50% socioeconomic status were compared (Fig. S13). Conversely, June-August born infants ($n = 55$) experience high food insecurity during this same developmental window as harvest stocks deplete. Here, no relationship between enrollment WAZ and 12-month LAZ was observed, independent of socioeconomic status (Fisher's Z-test $p = 0.025$; Figs. 4F, S13). This indicates that when food is scarce, enrollment weight has a limited influence on future growth. However, nicotinamide supplementation in infants born during this window did

restore a positive correlation between enrollment WAZ and 12-month LAZ ($n = 55$; $r = 0.34$, correlation $q = 0.046$; Fisher's Z-test $p = 0.044$; Fig. 4F). This demonstrates that supplementation may have ameliorated the impact of limited nicotinamide intake on growth.

Importantly, the enrollment status of the infant determined the efficacy of nicotinamide supplementation. As enrollment WAZ was observed to influence future growth, infants were stratified into enrollment WAZ tertiles. A two-way ANOVA and Tukey post-hoc tests were performed to investigate how birth season influences the impact of nicotinamide supplementation on infant growth (change in LAZ from enrollment to 18 months). Considering only infants in the upper tertile for enrollment WAZ (WAZ ≥ −0.13), a significant interaction was observed between birth season and nicotinamide supplementation in association with infant growth ($p = 0.024$, Supplementary Data. 19), which was not present in those with lower enrollment weights. Time

series analysis confirmed these findings, where nicotinamide supplementation in June-August born infants (i.e., those who enter increasing food insecurity at three to nine months of age) significantly reduced the drop in LAZ over 18 months compared to matched infants born in the same period not receiving supplementation ($p = 0.013$; supplemented $n = 19$, not supplemented $n = 26$; Fig. 4G). This improvement was not seen for December-February born infants (i.e., those experiencing decreasing food insecurity at three to nine months of age) with the same enrollment weights (supplemented $n = 35$, not supplemented $n = 28$; Fig. 4H). This highlights that the effectiveness of nutritional interventions can be influenced by the season of provision, shaped by environmental pressures and inherent metabolic states.

## Discussion

The developing metabolic phenotype of the mother-infant dyad strongly influences infant health and development, and its maturation is modulated by nutritional factors. Our findings show that in a unimodal harvest setting, the infant metabolome is dependent upon the month in which the infant is born, and this imprinting persists up to 18 months of life. We demonstrate that these patterns are shaped by food availability, including nutrient transfer from mother to infant, which is influenced by climatic pressures and seasonal variation in food insecurity. This included five circulating metabolites with developmental significance (choline, TMAO, EPA, hex2cer(d18:1/18:0), and cer(d18:2/14:0)), whose abundance varied based upon the time of year the infant was born, being consistent at both 12 and 18 months, despite fluctuations in food insecurity across these two sampling points. Interactome networks built on these longitudinal datasets neatly visualize the statistical relationships between environmental-maternal-infant features and ultimately phenotype (Fig. S14). For example, seasonal patterns in birth year rainfall were aligned with food insecurity at 12 months, which subsequently related to plasma TMAO in the infant, which was positively associated with cognitive outcomes at 18 months. Similarly, sequential relationships were observed between birth year rainfall, food insecurity at one month, nicotinamide-related metabolites in breastmilk, infant nicotinamide-availability, and growth at six months (Fig. S15).

Choline is critical for growth, muscle development, neurodevelopment, and cell membranes[27] and several metabolites related to choline metabolism were observed to be seasonal. The highest concentrations of 12-month free plasma choline were observed in infants born in December (mean 26.14 μM ± SD 8.29), during peak food insecurity, while June-born individuals exhibited the lowest concentrations (mean 19.16 μM ± SD 4.92). A Malawian study found infants aged 12–15 months had an average plasma choline of 14.6 μM[28] and similar values (16.3 μM) were noted at 12 months in a healthy, well-nourished Turkish cohort[29]. This indicates plasma choline abundance in infants born during low food insecurity is consistent with previously reported reference values. In contrast, plasma choline is elevated in those born during the rainy season when dietary choline intake is expected to be lowest. It is unclear if infants born during peak food insecurity are synthesizing greater amounts of choline endogenously or if they are metabolically programmed to preserve this important developmental molecule. From the urinary metabolic profiles, these infants were observed to excrete lower amounts of the choline-derived metabolites, betaine and DMG, at six months (when food insecurity is lowest), indicating reduced choline utilization. Betaine and DMG are generated through choline breakdown, driving the homocysteine-methionine cycle to synthesize the body's primary methylating agent $S$-adenosylmethionine. We have previously shown that stunted infants excrete less betaine and DMG compared to those without growth shortfalls, and that this is key throughout the first year of life[5,6]. Inherent reductions of choline processing in those exposed to early life nutritional deficiencies may reflect the programming of biochemical thrift and explain the limited effectiveness of choline-focused interventions[12].

Interestingly, seasonal variation was also observed in 18-month cognitive outcomes, where infants born in January-February had the greatest MDAT scores at 18-months. While this is a period of high food insecurity, these infants experience increasing food availability during their first six months of life following the bean harvest in March. This period is key for brain development, including axonal and dendritic arborization and increasing synaptic density[30,31]. Both plasma EPA and TMAO were positively associated with cognitive function and showed fixed birth-month-dependent seasonality when measured six months apart. TMAO is a microbial-host co-metabolite that arises from the microbial breakdown of dietary choline to trimethylamine before absorption from the gut and conversion to TMAO by host flavin-monooxygenase-3 enzymes. Plasma choline significantly cross-correlated with plasma TMAO at 12 months, with choline preceding TMAO by ten days. Therefore, the seasonality in TMAO likely reflects gut microbial activity in the infant, indicating birth season can shape the functional capacity of the establishing microbiota. We have previously shown that TMAO can reach the brain[32], and its abundance exhibits a developmental pattern. Moreover, Vuong et al. identified that the maternal microbiome can influence fetal neurodevelopment through its metabolic output, implicating TMAO as an influential signal with axonogenesis-promoting potential[33]. Furthermore, Hoyles et al. observed TMAO to enhance blood-brain barrier integrity and prevent cognitive performance loss following inflammatory insults[34]. EPA is an omega-3 PUFA obtained from oily fish or synthesized endogenously from alpha-linoleic acid breakdown. Despite this positive relationship between breastmilk and plasma EPA, the lack of association between breastmilk EPA at one- or five-months and cognitive outcomes aligns with the literature where several studies have noted mixed results on the effectiveness of omega-3 supplementation during lactation and cognitive outcomes of the infant[35]. These results, together with previous findings, emphasize the requirement for nutritional interventions targeting these metabolites in infants born during the dry season who enter postnatal periods of increasing food insecurity.

Seasonal variation in the breastmilk composition and transmission of seasonality from breastmilk to infant highlights the importance of considering the mother-infant dyad when designing nutritional interventions. We observed season-dependent variation in the HMO composition of both the breastmilk and six-month infant urine samples. Seasonally dependent HMO compositions have previously been observed in Bangladeshi, Canadian and Gambian cohorts[36–38], where greater HMO production was noted in Gambian mothers nursing during the dry season[36], a period of decreasing food insecurity. Through selective fermentation, HMOs are important for shaping the establishment of the intestinal microbiota with Bifidobacterium and Bacteroides species metabolizing these sugars to organic acids, reducing gut pH to provide resistance against pathobionts and providing an energy source for the host[39,40]. Seasonal changes in HMO composition may therefore modulate infant gut colonization trajectories, provide differing degrees of protection from infections and caloric recovery, and contribute to the variation seen in plasma TMAO.

It should be noted that other birth month effect modifiers on the metabolome are likely present. This includes factors that directly impart the seasonal metabolic associations, such as infections, and those that shape the magnitude of seasonal variation, such as socioeconomic status. For example, seasonal variation in cognitive outcomes and their association with plasma metabolites were only seen in those with the highest socioeconomic status (top 50%). Interestingly, sex differences were also observed where the male metabolome appears more sensitive to seasonal perturbations than the female. This aligns with literature reporting more distinct metabolic differences in adult males following periods of early life undernutrition compared to community controls[41,42], and a greater adverse response of the male fetus to stress during pregnancy and post-natal life[43,44].

Seasonal shortfalls were observed in nicotinamide-related metabolites in the infant and breastmilk, previously implicated with stunting, including NMND[5,19]. Urinary NMND is a biomarker of nicotinamide availability, whereby surplus nicotinamide is methylated and excreted as NMND. Nicotinamide is key for $NAD^+$ generation via the salvage pathway, and we and others have shown the tryptophan-niacin-$NAD^+$ pathway is perturbed with early life undernutrition and is related to growth[5,19]. This is due to reduced precursor availability (NR, nicotinamide, niacin, tryptophan) and immune-associated disruptions (induction of indoleamine 2,3-dioxygenase [IDO] activating the kynurenine pathway)[15,20]. While birth month does not appear to strongly influence LAZ, nicotinamide availability (reflected by seasonal breastmilk NR and nicotinic acid, and urinary NMND) may be one factor modulating growth. Here, we demonstrate that maternal nicotinamide supplementation was sufficient to increase breastmilk NR and subsequent infant NMND excretion, overcoming their seasonal variation and elevating nicotinamide availability. Despite the intervention not improving growth in the overall cohort[17], growth deficits were reduced in a specific subset of infants experiencing high food insecurity in their first nine months of life.

A limitation of this work is that the metabolomic patterns associated with the month of birth are likely to be driven by a myriad of intrinsic and extrinsic pre- and postnatal pressures, some of which are unaccounted for here. For example, variation in gut microbial composition, infection burden, immune responses and enteric dysfunction may also change depending on the time of year an infant is born and could contribute to these metabolomic differences[45-47]. Enteropathogenic infections are prevalent in this setting[48], with certain infections exhibiting seasonal variation, such as increasing Shigella during the rainy season[49]. Nevertheless, birth month represents a proxy for this combination of complex and interacting pressures. Seasonal fluctuations in post-harvest storage conditions, particularly variations in temperature and moisture[50], can alter the metabolic profiles of crops over time, as well as the location of production[51], potentially affecting nutrient and amino acid levels independently of food insecurity. This may alter the relationships between season and metabolite concentrations in other geographic settings. Lastly, contributions of nutritional or environmental factors that occurred during pregnancy prior to recruitment of the cohort were not captured and warrant further investigation.

This work highlights that in this setting, the season in which an infant is born has enduring effects on the maturing metabolome, an important factor shaping phenotypic development. This evidence supports the need to consider seasonality, both contemporary and around birth, when tailoring interventions targeting developmental consequences of early life undernutrition. These findings may also explain the limited success of nutritional and lifestyle interventions previously trialed in these settings[9-12]. For infants born in agriculturally dependent, low-resource communities, climate change will only exacerbate such nutritional variation and hence this will become an increasingly prominent consideration.

## Methods

The principle investigator of this project was Estomih, Mduma, Ph.D, Research Manager at Haydom Global Health Research Center, who led the local research effort, with a team that included local physicians, clinical research coordinators and field team members. The recruitment process began with the research team meeting with local ten-cell community leaders and educating them about the project's rationale and goals. Recruitment proceeded with local community healthcare workers, who identified to study field team members all prospective participants - mothers with known pregnancies or births. Recruitment was open to all families in the region regardless of race/ethnicity or social status. That said, the area around Haydom Lutheran Hospital is a low-resource setting, and all participants lived in poverty. The project goals focused on assessing potential interventions to reduce childhood stunting and increase cognitive development - issues clearly relevant to the local community. The results of the research have been shared locally, though the study demonstrated a lack of efficacy of the interventions tested, limiting any new benefit the community could experience.

## Study design

The Early Life Interventions for Childhood growth and development in Tanzania (ELICIT) study is a randomized 2 × 2 factorial, double-blind, placebo-controlled trial with the aim of improving early life growth using a nicotinamide and/or antimicrobial (azithromycin, nitazoxanide) intervention. This trial was based in the Haydom Global Health Research Center at Haydom Lutheran Hospital in Haydom, Tanzania. The primary outcome from the ELICIT study was 18-month mean LAZ, followed by the secondary outcomes of 18-month WAZ, MAZ, HCZ, hospitalization, all-cause mortality, and childhood illness. This manuscript performs a secondary explorative analysis of the cohort with the focus on biochemical seasonality. The detailed study design and methods of data collection as per the Consolidated Standards of Reporting Trials (CONSORT) guidelines have been previously published[15], alongside the baseline characteristics of the participants[17], the primary outcomes[14] and statistical analysis plan. Participants were enrolled into one of four treatment groups (1:1:1:1). This divided the individuals to receive either nicotinamide plus placebo ($n = 276$), antimicrobial plus placebo ($n = 277$), both nicotinamide and antimicrobial ($n = 267$), or both placebos ($n = 278$). In this manuscript, only infants from the placebo/placebo and nicotinamide/placebo arms of the study were considered. The seasonality analysis in this manuscript included placebo/placebo arm infants to observe seasonal patterns unaffected by the intervention. Only infants from the nicotinamide/placebo arm were included when considering the intervention to separate the individual intervention effects. The antimicrobial intervention was not considered in this manuscript as the biological samples were collected prior to the administration of the single-dose antimicrobials. Additionally, receipt of non-study antibiotics was common (54-86% infants).

## Study setting

Haydom is a rural area of north-central Tanzania, where the population of ~20,000 rely primarily on subsistence agriculture, determined by the areas unimodal crop cycle and rainy seasons. Here, food is available following the harvest of beans during the long rains (March-May) and of maize during the dry season (June-October). However, during the pre-harvest period (November-February), when the short rains and planting season occur, harvest stocks become depleted, and the diversity of the foods consumed is reduced. Families in this area typically do not have access to improved water sources and have a monthly income of less than 58,000 Tanzania shillings (~US$25)[14,17,43]. There is very low HIV seroprevalence in mothers of childbearing age in this population (<2%), and Haydom is not a high-risk malaria site due to the elevation (1700 m)[52]. The predominant dietary staple for participants was maize, accompanied by beans, which are typically consumed around the time of their harvest, and eggs. Sunflower oil was commonly accessible, while meat, chicken, dried fish, bananas, avocadoes, and mangoes were available but not consumed in significant quantities due to their high cost at local markets.

## Participants

Between September 5, 2017, and August 31, 2018, community health workers within the recruitment area identified pregnant women or those who had recently delivered. These women were then contacted by field team members at their homes to determine interest in the study. To be eligible to partake in the study, the following criteria were assessed: maternal age ≥ 18 years, child age ≤14 days, and the family's

intent to live no more than 25 km from Haydom Lutheran Hospital while the study was undertaken. Exclusion criteria for the study comprised of infant enrollment weight <1500 g, multiple gestation, significant birth defect or neonatal illness, and lack of intent to breastfeed. Infant sex was determined based on self-report.

### Randomization and blinding
Randomization was performed using permuted blocks (block size eight) and a reproducible seed by study author JPM, with these detailed contained at the research center. Blinding began with the intervention manufacturer, who provided both the intervention and placebo. An allocation code for the treatment and placebo was provided and kept by a non-study investigator in a sealed opaque envelope. Randomization block details were provided to field teams at enrollment to randomize participants. Blinding continued for the participants and investigators until study completion, and the primary outcome analysis was completed.

### Intervention procedures
Nicotinamide versus placebo (both manufactured by VITA-gen, New York, USA) was given to the mothers as a daily 250 mg tablet for the first six months of life. Throughout this time, trained community and study health workers provided mothers with breastfeeding support and counseling surrounding the guidelines of the World Health Organization (WHO) and the Tanzanian Ministry of Health of the benefits of exclusively breastfeeding for the first six months of life. From six-18 months of life, nicotinamide was directly supplemented to the infant by mouth as a daily 100 mg sachet. This sachet contained powder that was mixed into a small volume of age-appropriate food by the mothers. To assess compliance, pill and sachet counts were undertaken at each dose distribution time point (every two months), and each month, mothers were asked about the frequency of ingestion.

### Sample collection
The study design publication includes detailed information on sample and data collection, including the anthropometry, breastfeeding status, feeding questionnaires, childhood illness and treatments, vaccine reports, developmental assessments (including Malawi Developmental Assessment Tool [MDAT][21]), breastmilk, blood, and urine collection for metabolomic analysis[15]. Briefly, monthly home visits were performed from one to 18 months of life (±7 days). Economic and demographic information, such as the geographical area of residence (ward) and tribal affiliations, were provided by the mothers at the baseline visit. The socioeconomic status of the family was determined following a protocol previously designed from the Etiology, Risk Factors, and Interactions of Enteric Infections and Malnutrition and the Consequences for Child Health (MAL-ED) study based on improved water and sanitation, assets, maternal education, and household income (WAMI)[22]. At the following monthly visits, mothers were asked about breastfeeding status, childhood illness and treatment, and intervention compliance. Household food insecurity was also assessed at each monthly visit with the question "Since the last visit, has the mother worried that the household would not have enough food?". Any frequency of worry in response to this question was considered. At each three-month visit, anthropometric data were obtained for the child. Length was assessed by measuring board, weight by digital scales, and MUAC and head circumference with measuring tape.

Blood samples were collected at 12 (placebo $n = 199$; nicotinamide $n = 199$) and 18 (placebo $n = 199$; nicotinamide $n = 198$) months of life via phlebotomy, transported on ice to the laboratory to be processed into plasma, and stored at −80 °C for shipping to the United Kingdom for metabolomic analysis. Urine samples were collected into a sterile bag at six (placebo $n = 278$; nicotinamide $n = 276$), 12 (placebo $n = 270$; nicotinamide $n = 267$), and 18 months (placebo $n = 266$; nicotinamide

$n = 263$), before being transported on ice to the research site and then stored at −80 °C to be shipped to the United Kingdom for metabolomic analysis. Breastmilk was collected at one- and five-months postpartum (placebo $n = 292$; nicotinamide $n = 295$) at the midpoint of a feeding session. The area around the areola was cleaned using soap and water and rinsed with deionized water prior to sampling. Approximately 8 mL was expressed by hand into a sterile container, which was placed on ice for transportation to the laboratory. Here, the samples were aliquoted whilst being shielded from light and stored at −80 °C and shipped to the United States of America for metabolomic analysis.

Rainfall data was collected from the weather station in Haydom, including historic rainfall from July 2010-July 2018.

Full methodological details of the Malawi Developmental Assessment Tool (MDAT) are documented in the cognitive outcomes[53] and study design publications[15]. Briefly, this test has been designed and validated to test gross and fine motor, language, and social development domains in a culturally relevant manner for rural or lower-resource settings in sub-Saharan areas with good reliability (94–100%)[21]. For Haydom specifically, the MDAT has been adapted, piloted, and validated by the MAL-ED study team[54]. The assessment was delivered to participants in Swahili or Iraqw (the first language of most of the participants) by three trained field team members. To ensure accuracy in translation, the assessment was translated into Swahili and then back-translated into English by a second independent interpreter. For each task, individuals were scored either one (pass) or zero (not pass). They continued to complete tasks in each subtest until they completed six tasks consecutively with a score of zero. To monitor consistency, all assessments were video recorded and 10-20% of these were watched back each week by trained study team personnel.

### [1]H NMR spectroscopy and UPLC-MS-based metabolic profiling
Urinary metabolic profiles were measured by [1]H nuclear magnetic resonance (NMR) spectroscopy. Here, urine (630 ml) was combined with 70 ml of phosphate buffer solution (pH 7.4, 100 % D2O) containing 1 mM of the internal standard, 3-trimethylsilyl-1-[2,2,3,3-2H4] propionate (TSP). Samples were vortexed to mix, spun at 10,000 g, and the supernatant was transferred to 5 mm NMR tubes. All samples were measured with a 700 MHz Bruker NMR spectrometer equipped with a cryoprobe and refrigerated SampleJet autosampler maintained at 6 °C (Bruker Biospin GmbH, Rheinstetten, Germany). A pooled biological QC was prepared and analyzed repeatedly throughout the run to confirm reproducibility (Fig. S16). A standard one-dimensional solvent suppression pulse sequence (relaxation delay, 90° pulse, 4-ms delay, 90° pulse, mixing time, 90° pulse, acquire free induction decay) was used to measure each sample. Each spectrum was acquired with 32 scans, 4 dummy scans, 64,000 frequency domain points and a spectral window set to 20 ppm (parts per million). All spectra were automatically phase and baseline corrected and referenced to the TSP resonance at d0.0 in Topspin 3.2 (Bruker Biospin GmbH, Rheinstetten, Germany). The raw spectra were digitized, aligned, and normalized in Matlab (version 2018a, MathWorks Inc.) using the Imperial Metabolic Profiling and Chemometrics Toolbox (https://github.com/csmsoftware/IMPaCTS). Redundant spectral peaks (TSP, water, and urea) were excised, and the resulting spectra were manually aligned using a recursive segment-wise peak alignment method. Due to variation in dilution across urine samples, a probabilistic quotient normalization method was used to reduce these effects.

Plasma metabolic profiles were measured on a Waters Acquity Premier ultra-performance liquid chromatography system coupled to a Xevo TQ-XS mass spectrometer using the Biocrates MxP Quant 500 kit (Biocrates Life Sciences, Innsbruck, Austria)[55]. This kit was also used to generate breastmilk metabolic profiles using an ABSciex 5500QTRAP mass spectrometer. The analysis of small molecules (106

metabolites) was performed by LC-MS/MS and flow injection analysis for the lipids and hexoses (524 metabolites). This kit incorporates internal standards that are used for metabolite identification and quality control samples to assess reproducibility. This data was acquired following the manufacturer's instructions. The complete list of metabolites included is documented in the kit specification[56], and the quantification was achieved using a 7-point standard curve of internal standards. Here, 10 ml of thawed samples were added to a 96-well plate containing inserts of internal standards supplied by Biocrates and dried using nitrogen flow. Samples were incubated with 5% phenyl isothiocyanate (PITC), dried, shaken following the addition of 5 mM ammonium acetate in methanol, before being transferred to separate 96-well plates diluted with either pure water or FIA solvent. Metabolites were measured in positive and negative ionization modes. The LC-MS/MS method involved the injection of 5 μl of extract into the Biocrates MxP Quant 500 kit UHPLC column at 50 °C using a 5.8 min solvent gradient with 0.2 % formic acid in water and 0.2 % formic acid in acetonitrile. Secondly, the FIA-MS/MS method involved the injection of 20 μl of extracts. The raw data from the UPLC-MS and FIA analyses were processed using the Biocrates MetIDQTM software (version Oxygen-DB110-302305) in accordance with the manufacturer's protocol. To minimize technical variation across the different plates, batch correction was performed using standard quality control samples (QC2) repeated across ($n = 4$) each plate (Fig. S16).

Breastmilk vitamin and HMO profiles were also generated. For the vitamin profiles, internal standards were added to 100 ml of whole human milk samples, mixed, and centrifuged at 10,000 x $g$ for 10 min at 4 °C to defat the sample. A 50 ml aliquot of whey was taken, and the proteins were precipitated with methanol. After another centrifugation step, the supernatant was transferred into a new glass vial, evaporated to dryness, reconstituted, and filtered. Analysis was performed using a SCIEX Exion-AD UHPLC coupled to a SCIEX 6500 + QTRAP mass spectrometer (SCIEX, Framingham, MA, USA), using a Phenomenex NX-C18 column, 250 × 3 mm, 3 μm, and a gradient of an aqueous solution of 10 mM ammonium formate + 0.05% ammonium hydroxide (A) and acetonitrile (B) from 99% to 5% A in a 12 min run. Quantification was carried out by ratio responses using isotope-stable internal standards and a 9-point standard curve. Additionally, the HMO profiles were generated for a subset of participants (placebo $n = 54$; nicotinamide $n = 48$) from the breastmilk samples. These were generated on a high-performance LC system with fluorescence detection. Firstly, raffinose was added to 40 μl of breastmilk as an internal standard to facilitate quantification. A high-throughput solid-phase extraction was used to isolate the HMOs, involving the removal of lipids, proteins, and lactose. The resultant HMOs were fluorescently labeled for HPLC-FLD detection based upon retention times and mass spectrometry.

Variables with greater than 80 % missing values (i.e., those below the limit of quantification) across the sample set were excluded. Outliers were identified using Grubbs' test of log-transformed data and removed.

## Statistical analysis and data visualization

Cosinor analysis was performed in MATLAB (version R2022b, MathWorks Inc.; Statistics and Machine Learning Toolbox; Optimization Toolbox). Analyses were applied to metabolomic datasets (UPLC-MS-based plasma and breastmilk profiles, and urinary $^1$H NMR profiles), as well as metadata relating to the individuals (anthropometric measures at different sampling points, cognitive outcomes) and environmental variables (Haydom rainfall data and food insecurity reports) across the birth months of participants. For breastmilk data, cosinor analysis was applied across sampling months. For the urinary $^1$H NMR data, the cosinor approach was applied to each datapoint in the urinary $^1$H NMR profiles. Metabolite peaks with datapoints identified to significantly fit cosine distributions were integrated to obtain the relative abundance

of that metabolite. Only those metabolites whose peak integral also followed a significant cosine distribution were included in further analysis. As food insecurity was recorded from each mother monthly over their 18-month study period, the mean food insecurity scores were calculated across all mothers for each 10-day period, and any response of worry was included.

A cosine wave with a sloping mesor was fitted to the data using non-linear curve fitting solved by least squares, and statistically compared to H0 of solely a sloping line model using an extra-sums-of-squares $F$-test. All $p$-values were corrected for multiplicity of testing according to Benjamini-Hochberg. Features with a $q < 0.15$ at at least one sampling time point were considered significant cosine fits and included in further analysis. This threshold was selected due to the large number of metabolites measured and the covariation that exists across the metabolomic dataset through shared metabolic pathways or functions. The analyses were repeated using residualized features according to a linear model including potential covariates (enrollment weight-for-age z-score, socioeconomic status, number of months of exclusive breastfeeding, and maternal factors (weight, height, age)) and stratified by sex. As the metabolomic-lipidomic and HMO profiles were only measured in a subset of samples, seasonal differences were investigated using one-way ANOVA with Tukey's HSD post-hoc for normally distributed residuals, or Kruskal-Wallis with pairwise Wilcoxon rank-sum tests performed in R (version 4.2.1; stats package[57]. Features with a $q < 0.15$ were considered significant.

PERMANOVA analysis assessed variance in the metabolomes of participants by birth season using metabolites identified as following a 12-month cosine distribution using the vegan package (version 2.6.10) in R[58]. This was performed on scaled data with the LOD replaced with the minimum value divided by √2. An *adonis* $p$-value was produced following 1000 permutations, where $p < 0.05$ indicated that participants were clustered significantly more with others from the same birth season. PLS-DA models were used to confirm the classification of participants into three-month birth windows by the metabolites identified as following a 12-month cosine distribution at six, 12, and 18 months and generate VIP scores. For each model, the X matrix was composed of the concentrations of the metabolites (scaled data with the LOD replaced with the minimum value divided by √2), whilst birth season (September-November versus March-May; December-February versus June-August) was used as the predictive Y component. PLS-DA models were performed using the mixOmics (3.21) package in R[59]. A seven-fold cross-validation approach was used to assess the predictive capacity of each model, while permutation testing demonstrated the model's validity (1000 permutations). Area under curve (AUC) and variable importance (VIP) scores were generated for each model.

Cross-correlation analysis was also performed in MATLAB (version R2022b, MathWorks Inc.; Statistics and Machine Learning Toolbox, Bioinformatics Toolbox, Signal Processing Toolbox). Metabolite concentrations were z-scored within each metabolite, and cross-correlated with reported food insecurity to find the optimal lag (largest correlation) and associated $R^2$ and $p$-value. All $p$-values were corrected for multiplicity of testing according to Benjamini-Hochberg, and $q < 0.05$ were considered statistically significant correlations. A stricter threshold was used for cross-correlation analysis due to the independence of the variables tested. For the metabolomic-lipidomic and HMO breastmilk subset, Wilcoxon tests were used to detect metabolites that differed by reported food insecurity (any vs no) by mothers at the time of sampling. An $q < 0.05$ was also used to detect significance in this analysis. Analyses were performed in R using stats package[57].

Short time series analysis was performed using the Short Asynchronous Time-series Analysis (santaR) package (version 1.2.3)[60] in R to determine differences in growth trajectories between the intervention groups, and urinary NMND excretion tertiles. This method uses a smoothing spline functional model where six splines were selected when modeling anthropometric change over seven time points.

Other analyses were performed in R version 4.2.1 primarily using the following packages: ggplot2 version 3.4.1 for plotting[61], dplyr version 1.0.10 for data manipulation[62], and stats[57] for statistical analyses. The exact number of samples used to calculate a statistical difference is indicated in each figure legend. Shapiro-Wilk's tests were used to assess the normality of the data. For normally distributed data, Student's T tests and Pearson's correlations were used. Conversely, Spearman's Rank correlations and Wilcoxon tests were used when analyzing skewed data. The false discovery rate was corrected using the Benjamini-Hochberg procedure to correct $p$-values for multiple testing. Features with a $q < 0.05$ were considered statistically significant in the analysis unless stated previously. A Fisher's Z transformation was applied to the correlation coefficients, and a $Z$-test was used to determine statistical difference. Growth data were analyzed using a Two-Way Analysis of Variance (ANOVA) with birth season and intervention group as independent variables. Adobe Illustrator was used to finalize the figures.

## Inclusion and ethics

The study, and transfer of data and materials, were approved and was closely followed by all local research regulatory agencies, including the Tanzanian National Institute for Medical Research (NIMR, reference number HQ/R.8a/Vol.IX/2424), the University of Virginia Health Sciences Research Institutional Review Board (HSR-IRB, reference #19465), and Tanzanian Food and Drug Administration (reference number TFDA0017/CTR/0005/02). The study also received ethical approval from the University of Southampton Research Ethics Committee (reference number 61337.A1). Study oversight was provided by FHI360 (North Carolina, United States of America). This study was registered in ClinicalTrials.gov (NCT03268902) prior to the start of enrollment. Mothers gave written informed consent for themselves and their infant to participate in any intervention arm either during pregnancy or at the time of enrollment.

## Reporting summary

Further information on research design is available in the Nature Portfolio Reporting Summary linked to this article.

## Data availability

Data supporting the findings of this study are available in the accompanying Source Data files or in the MetaboLights repository under accession number MTBLS13213. Source data are provided with this paper.

## Code availability

No code is available related to these analyses.

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

## Acknowledgments

This study was funded by the Bill & Melinda Gates Foundation, OPP1141342, JRS, RS, MDD, JPM, EM. URL: https://www.gatesfoundation.org/. The funder assisted with the design of the study but was not involved in the data collection and analysis, decision to publish, or preparation of the manuscript. Nicotinamide and a corresponding placebo were provided by Vita-gen. who had no role in study design, data collection and analysis, decision to publish, or preparation of the manuscript.

## Author contributions

Conceptualization: J.R.S., M.D.D., R.S., E.M., and J.P.M. Formal analysis: E.A.W. Investigation: E.M., L.B., S.J., P.M., J.P.M., R.S., M.D.D., G.P., M.P., J.R.S., D.H., and L.A. Methodology: E.A.W., J.R.S., and D.vd.V Project administration and funding acquisition: M.D.D., J.R.S., and E.M. Software: D.vd.V. and E.A.W. Visualization: E.A.W. Writing – original draft: E.A.W. and J.R.S. Writing – review and editing: all authors.

## Competing interests

The authors declare no competing interests.

## Additional information

[1]School of Human Development and Health, Faculty of Medicine, University of Southampton, Southampton, UK. [2]Department of Nutrition, University of California, Davis, CA, USA. [3]USDA, ARS-Western Human Nutrition Research Center, Davis, CA, USA. [4]Chronobiology section, Faculty of Health and Medical Sciences, University of Surrey, Guildford, UK. [5]Department of Medical Parasitology and Infection Biology, Swiss Tropical and Public Health Institute, Allschwil, Switzerland. [6]University of Basel, Basel, Switzerland. [7]Department of Medicine, University of Virginia, Charlottesville, VA, USA. [8]Haydom Global Health Research Centre, Haydom Lutheran Hospital, Haydom, Tanzania. [9]Department of Paediatrics, University of Virginia, Charlottesville, VA, USA.
✉e-mail: J.Swann@soton.ac.uk

