## [Transparent Peer Review file · Nature Communications]

Birth season shapes the infant metabolome and development in Tanzania: a secondary explorative analysis of The Early Life Interventions for Childhood growth and development in Tanzania (ELICIT) trial

Corresponding Author: Professor Jonathan Swann

Version 0:

Reviewer comments:

Reviewer #1

(Remarks to the Author)

While the idea of the study is of interest, there are several issues that have not been properly accounted for. The clinical cohort is not very clearly described and while the description is lengthy it is not easy to find information of the number of samples for each sample type, and there is no clinical data stated for the infants or their mothers, nor how many participants there were for each season, breastfeeding at the different timepoints etc.. Overall, the sample size in placebo part of the study was quite small, so the number of subjects for each month must have been relatively low. A table summarising this would be good to include. Supplementary files were not available, making a detailed investigation difficult. The statistical analyses are not properly described and there is not sufficient description of the possible confounding factors and how these were taken into consideration in the statistical analyses. Only enrolment weight and length of breastfeeding are mentioned, but other factors, such as the weight at sampling, infant sex, dietary factors, maternal factors such as BMI and age are very relevant in relation to metabolic profiles. There is no data on how the breastfeeding impacted the results. There is no quality control of the metabolomics data or description of the reliability of the metabolite identification. The authors state that the birth month showed significant association with specific metabolites. However, the results could be potentially explained by the month the samples were collected, and the food-intake/availability at that time. This is directly linked with the birth month, and thus the food intake on the sampling point. There was a short comment that there was a one sampling-month related metabolite. This is not sufficient, and a more comprehensive analysis/description should be added. How was the situation with the urine metabolome? There should be more discussion of this on the manuscript. This is the major issue on this study.

Related to urine samples, the results showed several metabolites stated to be linked with breast milk. How many infants were still breastfed at the two later sampling points?

There is some data given from the breast milk samples, however, the duration of the breastfeeding, and its impact on metabolic profiles is not described and as the sampling time of the breast milk did not fully match with the sampling of blood and urine, there are several possible issues that should be better explored in the study. Also the BM samples were grouped by birth month while it would be much more logic to group them by season of sample collection. Maternal diet and food availability is likely to have a major impact of the BM composition.

The figure texts lack sufficient description; none of the figures explains what are the panels A, B, C etc.

Reviewer #2

(Remarks to the Author)

Overall Comments:

This study sought to characterize the effects of birth month on the infant metabolome through the first 18 months of life in Tanzanian infants. Other recent work in the field has shown that birth month/season leaves epigenetic imprints on infants and that seasonal weather changes can alter intervention effectiveness, food availability, and pathogen exposure. The authors used a multidisciplinary approach that compared MS and NMR identification and quantification of urinary and

plasma metabolites with breastmilk metabolites, infant anthropometry, and infant language scores, while also incorporating rainfall data, knowledge of local growing seasons, and survey data about perceived food insecurity during the birth month and at the time of sampling. The authors show evidence that birth season has long-lasting effects on important aspects of the infant metabolome; that these changes seem to correlate with season-related changes in food insecurity in the first six months of life; and that a subset of metabolites are correlated with differences in language development at 12 and 18 months. Importantly, the authors explore the efficacy of a nicotinamide intervention to alter birth-month-related deficits in this crucial vitamin and find that supplementation makes a significant corrective impact on WAZ at enrollment to LAZ at 12 months in the group most impacted by birth-season-associated food insecurity. Overall, this paper provides valuable insights into birth-season-related effects on infant undernutrition, which can be applied by those planning future interventions in low and middle income countries to maximize intervention effectiveness. I offer the following as suggestions to improve the manuscript.

Major Comments:

Introduction: In the first paragraph, it would be valuable to briefly mention other potential sources of birth month impacts on infant health and development including pathogen exposure/enteric dysfunction responses to rainfall, before getting into why and how this study chooses to look at the impacts of food insecurity. Though covered in the discussion, these additional sources are important enough to be considered throughout the paper. It would also underscore the importance of this study to briefly mention the potential impacts of climate change on seasonal variation in rainfall and food insecurity. Likewise, the second paragraph should include a discussion of the seasonal patterns in pathogen infection in Haydom. In Line 61, it would be helpful to include an overview of the arms in the trial and the impact of the interventions on the primary and/or secondary outcomes of the trial. This would improve the rationale or set up of the third paragraph describing effect measure modification.

Page 2, Lines 87-90: Please provide an overview of the ELICIT trial arms. Include details on the trial set up (i.e., a 2x2 factorial design, administration of antimicrobials (azithromycin and nitazoxanide, randomised together) and nicotinamide). Provide a rationale for why this secondary analysis only used a subset of two arms. Also, provide the total N from these two arms.

Page 6, Lines 204-205: It is unclear what this hypothesized relationship is based on - perhaps missing reference or reference to supplementary data - "These were all highest in infants born during peak food insecurity (November-February) and may be associated with maternal-infant wasting around the time of delivery."

Figure 3: Panel labels effective here; would be nice to see these added to other figures. For Panel F though, the subtitle feels misleading since numbers represent phase lag in days rather than any seasonal variation – consider changing the label to "Peak phase season variation in breastmilk vitamins"

Figure 4: Similarity in color between upper and lower tertile and the other key (control and nicotinamide) makes it difficult to distinguish them. Would also suggest moving control and nicotinamide key somewhere more central so it resembles the other legends or making it larger/more obvious. Would recommend rearranging panels so that axes and deeper similarities align more. Ex: Panels A,G,H would be easier to understand in a row; Panels B/C, D/E would make more sense stacked atop one another.

Discussion: A thorough discussion of birth month effect modification by different factors (by sex, SES, enrolment WAZ etc. as mentioned in results pg 4 line 150-153) is warranted here. Since these differential effects have major implications for intervention planning, they are important to interpret and discuss.

Page 12, Lines 433-435: This could be expanded to include more specifics, especially regarding effects of rainfall on enteropathogen exposure and potentially EED. This underscores the importance of targeted interventions as children in the crucial developmental phase that encompasses the first six months of life may be facing the dual challenges of increasing food insecurity and increased exposure to environmental pathogens during the rainy season.

Minor Comments:

Page 1, Line 27-28: "contributions of this to these adverse phenotypic outcomes" feels awkward due to unnecessary repetition of "this". Consider changing to "it is unclear if this imprints on the developing metabolome and contributes to adverse phenotypic outcomes"

Page 1, Line 30: consider changing "persisting up to 18 months of life" to "persisting up to at least 18 months of life" to clarify that this finding is due to length of study.

Page 2, Line 48: consider deleting "its influence".

Page 2, Line 72-73: should reword "the tryptophan-niacin-NAD⁺ pathway is perturbed with early life undernutrition and is related to growth" to "the tryptophan-niacin-NAD⁺ pathway is related to growth and perturbed with early life undernutrition" to simplify and clarify importance.

Page 13, Lines 474-488: Figure 3 caption missing extensive list of acronyms and abbreviations.

Reviewer #3

(Remarks to the Author)

This is an interesting and well-conducted study, showing that seasonal availability of food affects the infant metabolome in a low-income context, with potential implications for health and development.

I request the authors address the following comments, which I hope can strengthen the paper:

The study does not state any clear hypotheses and there are no sample size calculations. Could the authors clarify what aspects of the analysis plan were defined a priori, including e.g. decisions on seasonal bins, length of phase lag and stratification analyses? The obvious concern here is the risk of 'false positives', given the multiple outcomes from metabolomics conducted on plasma and urine samples. The authors attempt to address this using false discovery rate corrections. I am not qualified to comment in detail on the statistics, but do request the authors justify the decision to apply FDR corrections at $p < 0.25$ for some analyses as reported at lines 454 and 484, rather than the 'typical' $p < 0.05$.

The decision to look only at the placebo/placebo arm and nicotinamide/placebo arms of the study (line 543) requires justification.

Full details on sample collection and analysis would be good, bearing in mind that sample handling is so important for metabolomics. The level of detail is insufficient for reproducibility.

The authors should discuss why seasonal relationships in plasma and urine metabolite concentrations are seen only in males but not in females (line 151).

Looking at the seasonal patterns in metabolite concentrations (Figure 1) – the charts are plotted by month, from September to August. For several metabolites including choline, the concentrations in September and August are substantially different with no overlap in the 95% CI, yet these are adjacent months. How can this be explained? To me, this points towards other, non-seasonal drivers, possibly in combination with large sampling or measurement error when sample sizes are subset by month?

The study assumes a pathway from season to food insecurity to infant metabolome. There are, however, other potential pathways between season and infant metabolome that merit consideration. For example, moisture and temperature can affect cereal grain metabolism post-harvest. The aspartate data stand out to me, given associations between climate conditions during the growing of cereals and the free asparagine concentration in the resultant crop, e.g. Curtis et al., *Journal of Agricultural and Food Chemistry* 2009 57:1013-1021 DOI: 10.1021/jf8031292

Also, there may be seasonal variation in the geographic source of food items, e.g. more home-produced food in months after harvest, and more purchased food (likely grown further from the home) in the 'lean' season. Asparagine (and potentially other) amino acid concentrations, and other chemical entities including mineral micronutrients, are likely to vary depending on the location of production, including due to soil factors (e.g. Gashu et al., *Nature*. 2021;594:71-6), hence there will be seasonal variability in nutrient intakes independent of experience of food insecurity, and potentially important spatial variability in associations between season of birth, nutrient intakes and the infant metabolome. At line 436, the authors suggest that birth month acts as a proxy for these combination of seasonal drivers – however, these diverse pathways could lead to very different relationships between season and metabolite concentrations in other geographic settings.

The authors examined seasonal associations between metabolites and cognitive outcomes. At line 377, the authors suggest the observed associations demonstrate developmental relevance of metabolic seasonality. I think this statement is not sufficiently evidenced by the findings of this exploratory analysis. There are many pathways between season of birth and cognitive development, and many potential confounders for the association between metabolite concentrations and cognitive test scores. For example, labour patterns are likely to be seasonal which may affect breastfeeding patterns, and evidence from other settings including the UK shows associations between season of birth and cognitive development outcomes with hypothesised pathways including seasonal variability in maternal vitamin D status and opportunities for social interactions. And, of course, seasonal variability in consumption of food groups including fruits and vegetables could influence the infant metabolome and cognitive development without a direct interaction between the latter two. The authors do note limitations at line 431, but the inherent limitations of the study design are given insufficient weight and I suggest the authors use more cautious language in suggesting a direct relationship between metabolites and cognitive outcomes.

According to Figure 3A, reported food insecurity in the lean season was greater in 2017 than 2018 and probably 2019. Was 2017 a particularly bad year? The rainfall data suggest two false starts to the rains and a late, intense and short rainy period. Could the authors comment on this, and any potential implications for the wider external validity of the seasonal associations observed?

I am not convinced that you would deliver any common public health interventions differently on the basis of this study, when it is already well-known that the hungry season brings food security and nutrition-related challenges. Perhaps the authors can explain which aspects of common interventions or which particular interventions might be adapted or tailored for seasonality, where this does not already occur? To my knowledge, supplementation with choline or nicotinamide are not common public health interventions. However, the study could be used to inform the design of relevant future studies including monitoring and evaluation studies, including in terms of sampling timings.

The inclusion and ethics section lacks detail. Please could the authors provide information on the ethical review committees

and relevant application reference numbers? Also, please clarify whether the current study, i.e. analysis of infant metabolites, was included in the main ELICIT study protocol as reviewed and approved by the relevant ethics committees, or whether the current study operated under separate protocols and approvals? This section should also specify information on data and material transfer agreements.

The Supplementary Tables were not available in the reviewer materials and I am not able to comment on the adequacy of the authors' data sharing plans.

Minor comments

Line 214. This should be seasonal variation in, not via, breastmilk
Figure S1. There is a typo in section C, i.e. 'individuals'

Version 1:

Reviewer comments:

Reviewer #1

(Remarks to the Author)

While the manuscript has improved, there are now several parts on the new data that is making the interpretation more confusing, and it is still not fully clear that the birth month is the driving factor is many of the observations listed here. The main concerns are that of the seven metabolites that after adjustment are linked with birth month (choline, TMAO, EPA, methionine sulfate, cer(d18:2/14:0), TG 20:0_34:1 and hex2cer(d18:1/18:0) most of them are also linked with seasonal variation (choline, TMAO, EPA, cer(d18:2/14:0), TG 20:0_34:1 and hex2cer(d18:1/18:0). This is not clearly discussed in the manuscript. Which one is the main driving force, the birth month or the season? In addition, the associations seem to be rather weak, as FDR correction of 0.15 was used, normally one would use FDR= 0.05 or in some case, FDR of 0.1. How would the results look if more stringent FDR would be used? Overall, it would be good to have a similar plot that is shown in Fig1D for choline for all metabolites that were listed significant, showing also the error range of measurements, both related to the birth month and for month of sampling.

The circannual patterns were adjusted with sex, and after that, no significant patterns were observed in female infants. Why was not the birth month related analyses stratified by sex?

Urine metabolomics, why only the impact of birth months has been investigated, and the impact of sampling month has not been done in a similar manner than was done for plasma metabolomics?

PCA model shown in Fig S3, is the sampling month considered in the model? How the model would look if the grouping would be done according to the sampling month, rather than birth month?

Figure 1 E-J are not explained in the text. Fig 1 E is not very informative.

Line 109-110: The authors have added information that large part of the cohort has been getting antibiotics. Use of antibiotics can have a significant impact on metabolome, was this considered in the analyses? Is the use of antibiotics related to season?

There is still no information of the quality control. Just stating that the analyses were using standard manufacturers protocols does not ensure robust results, neither does the use of quality control samples. The LC-MS is known to be prone to analytical variability, even when standard protocols are being applied. As the QC samples have been used, it should be straightforward to give the data on the QC analyses. Statement that a batch correction was used, for quite a low number of analyses, raises concerns of the quality of the analyses.

Reviewer #2

(Remarks to the Author)

Thank you for comprehensively addressing my comments. A few additional comments below.

Minor Comments:

Line 98-100: "Here, data from the placebo group (n = 278) and infants receiving nicotinamide (n = 276, breastfeeding mothers provided 250 mg daily tablets for six months post-delivery, followed by direct infant supplementation (100 mg sachets) from six to 18 months) was studied."

The layered parenthetical is obscure and difficult to understand. The additional details about how the nicotinamide was introduced are important and deserve their own sentence.

Line 56-57: "As climate change is a major contributor to increasing food insecurity, understanding and preventing its detrimental impact on early life undernutrition remains crucial."

This sentence would benefit from a more thorough transition, clarifying how seasonality is related to anticipated changes in

climate, especially regarding droughts.

Reviewer #3

(Remarks to the Author)

Thank you for considering and addressing my comments. I am happy with the changes made.

Version 2:

Reviewer comments:

Reviewer #4

(Remarks to the Author)

The authors present a comprehensive analysis of how birth season shapes the infant metabolome and developmental outcomes in a large Tanzanian cohort. This revision has addressed the major concerns raised previously and substantially strengthened the manuscript in response to Reviewer 1's comments. The distinction between birth-month and sampling-month dependent metabolites is presented with schematics and supplementary figures. Additional analyses and explanations (stratification by sex, inclusion of antibiotic exposure, QC plots, and FDR thresholds) directly address the earlier points.

Reviewer 1 rightly noted that FDR thresholds of 0.05 or 0.01 are more conventional and stringent. The use of an FDR threshold of 0.15 is more lenient, but given the exploratory nature of the study, the consistent seasonal patterns observed for several metabolites (e.g., choline, TMAO, EPA, ceramides) across multiple time points, and the transparent reporting of stricter cut-offs in the supplement, this approach is acceptable.

The manuscript is much stronger, though a few minor refinements would improve clarity:

- Emphasize that "birth month" functions as a proxy for seasonal exposures rather than a causal factor.
- Make explicit in the main text where stricter FDR thresholds still support the robustness of key findings.

Reviewer #5

(Remarks to the Author)

REVIEWER COMMENTS

Reviewer #1 (Remarks to the Author):

- The clinical cohort is not very clearly described and while the description is lengthy it is not easy to find information of the number of samples for each sample type, and there is no clinical data stated for the infants or they mothers, nor how many participants there were for each season, breastfeeding at the different timepoints etc.. Overall, the sample size in placebo part of the study was quite small, so the number of subjects for each month must have been relatively low. A table summarising this would be good to include.
 - Sadly, it appears that the supplementary tables were not available to the reviewers. This seems to be confirmed by the reviewer "Supplementary files were not available, making a detailed investigation difficult". Supplementary table 1 details participant characteristics and numbers split into birth seasons. Following this suggestion, we have now added sample numbers per birth season to this table. As can be seen from the table, sample numbers were sufficient for the described analyses given that we detect interesting seasonal changes with several replicated over multiple time points. Furthermore, by adjusting for multiple testing we have limited the risk of false positives.

Supplementary Table ST1: Baseline characteristics of the cohort by intervention arm and birth season.

Birth Season	Placebo Arm				Nicotinamide Arm			
	Sep-Nov	Dec-Feb	Mar-May	Jun-Aug	Sep-Nov	Dec-Feb	Mar-May	Jun-Aug
Total individuals	43	116	83	55	42	114	87	55
Samples (n)								
Plasma								
12 month	22	81	61	35	22	74	68	35
18 month	22	80	61	36	22	74	68	34
Urine								
6 month	39	112	75	52	40	105	80	51
12 month	39	104	70	48	39	101	77	50
18 month	36	105	71	44	38	102	79	47
Breastmilk B vitamins								
1 month	39	115	82	52	41	112	86	55
5 month	41	116	82	53	40	108	85	53
Breastmilk metabolome and HMOs								
1 month	4	14	24	12	3	6	29	10
5 month	4	14	24	12	3	6	29	10
Sociodemographics								
Females	18 (41.86)	63 (54.31)	40 (48.19)	19 (34.55)	23 (54.76)	53 (46.49)	39 (44.83)	30 (54.55)
Firstborn child	36 (83.72)	92 (79.31)	63 (75.9)	44 (80)	33 (78.57)	103 (90.35)	66 (75.86)	48 (87.27)
Number of children	4 ± 2.71	3.97 ± 2.69	3.8 ± 2.44	3.83 ± 2.53	4.02 ± 2.22	4.14 ± 2.29	3.8 ± 2.51	3.89 ± 2.16
Hospital birth	20 (46.51)	62 (53.45)	55 (66.27)	30 (54.55)	20 (47.62)	50 (43.86)	47 (54.02)	28 (50.91)
Maternal age	27.26 ± 7.25	27.81 ± 7.23	27.37 ± 7.05	28.36 ± 6.97	27.71 ± 6.12	27.39 ± 5.66	27.76 ± 6.58	28.11 ± 6.13
Mothers with >= 7 years education	32 (74.42)	93 (80.17)	68 (81.93)	43 (78.18)	29 (69.05)	79 (69.3)	67 (77.01)	43 (78.18)
Monthly income (1,000 Tanzanian shillings)	50.66 ± 77.88	46.9 ± 43.48	52.78 ± 69.52	54.81 ± 85.46	60.08 ± 62.72	48.12 ± 40.44	40.12 ± 46.63	52.73 ± 49.09
Risk factors								
Exclusive breastfeeding in the first month	42 (97.67)	116 (100)	83 (100)	54 (98.18)	41 (97.62)	113 (99.12)	86 (98.85)	55 (100)
Routine treatment of drinking water	4 (9.3)	19 (16.38)	14 (16.87)	4 (7.27)	6 (14.29)	19 (16.67)	5 (5.75)	6 (10.91)
Drinking water > 10 minutes from home	32 (74.42)	98 (84.48)	69 (83.13)	43 (78.18)	35 (83.33)	100 (87.72)	64 (73.56)	44 (80)
Crowding	9 (20.93)	17 (14.66)	19 (22.89)	9 (16.36)	10 (23.81)	18 (15.79)	17 (19.54)	15 (27.27)
Water and sanitation, Assets, Maternal education, and household income index score (median, interquartile range)	0.30, 0.14	0.33, 0.18	0.31, 0.14	0.36, 0.18	0.33, 0.16	0.30, 0.17	0.30, 0.16	0.30, 0.16
Household assets								
Mattress	25 (58.14)	69 (59.48)	50 (60.24)	39 (70.91)	20 (47.62)	56 (49.12)	45 (51.72)	37 (67.27)
Table	18 (41.86)	56 (48.28)	38 (45.78)	32 (58.18)	19 (45.24)	37 (32.46)	28 (32.18)	27 (49.09)
Bench	39 (90.7)	102 (87.93)	73 (87.95)	49 (89.09)	37 (88.1)	94 (82.46)	65 (74.71)	46 (83.64)
Separate kitchen	27 (62.79)	67 (57.76)	59 (71.08)	38 (69.09)	28 (66.67)	66 (57.89)	53 (60.92)	37 (67.27)
Refrigerator	0 (0)	0 (0)	1 (1.2)	1 (1.82)	0 (0)	0 (0)	0 (0)	0 (0)
Television	3 (6.98)	5 (4.31)	5 (6.02)	6 (10.91)	7 (16.67)	9 (7.89)	4 (4.6)	4 (7.27)
Mobile phone	32 (74.42)	94 (81.03)	71 (85.54)	47 (85.45)	36 (85.71)	84 (73.68)	73 (83.91)	49 (89.09)
Family bank account	3 (6.98)	9 (7.76)	6 (7.23)	3 (5.45)	2 (4.76)	3 (2.63)	1 (1.15)	3 (5.45)
Anthropometry								
Enrollment weight-for-age z-score	-0.76 ± 1	-0.72 ± 0.93	-0.51 ± 0.98	-0.28 ± 0.93	-0.65 ± 0.97	-0.78 ± 1.08	-0.55 ± 0.9	-0.4 ± 1.02
Enrollment length-for-age z-score	-0.58 ± 1.14	-0.66 ± 0.96	-0.85 ± 1.00	-0.84 ± 0.87	-0.59 ± 0.98	-0.73 ± 1.06	-0.85 ± 1.02	-0.84 ± 1.14
Enrollment head-circumference-for-age z-score	0.03 ± 1.18	-0.12 ± 0.96	-0.06 ± 1.09	0 ± 0.92	0.22 ± 1.14	-0.06 ± 1.03	0.06 ± 0.89	0.26 ± 0.91

Mean ± SD is shown for continuous variables and the number (percentage) for dichotomous variables unless otherwise stated.

2.

a. The statistical analyses are not properly described

- All the statistical analyses performed in this work are clearly described in the methods section (Page 20-21; line 775-843). Following this suggestion, we have now added a sentence to main results section to clearly explain the cosinor analysis and improve interpretation (Page 3; line 118-120).
- *“Cosinor analysis (20) was used to identify molecules whose concentrations followed a seasonal waveform across the population based upon the infant’s month of birth (Fig.1A-C). A cosine wave with a sloping mesor was fitted to the data using non-linear curve fitting solved by least squares, and statistically compared to H0 of solely a sloping line model using an extra-sums-of-squares F-test.”*

b. there is not sufficient description of the possible con-founding factors and how these were taken into consideration in the statistical analyses. Only enrolment weight and length of breastfeeding are mentioned, but other factors, such as the weight at sampling, infant sex, dietary factors, maternal factors such as BMI and age are very relevant in relation to metabolic profiles. There is no data on how the breastfeeding impacted the results.

- In this work, cosinor analysis was performed on both unadjusted and adjusted metabolomes. Previously, the analysis included enrolment weight as a covariate to adjust for prenatal exposures, and length of breastfeeding to account for differences in dietary factors. Additionally, to account for infant sex, the analysis was performed in a sex stratified manner.
- We thank the reviewer for their suggestion, and we have now additionally included maternal height, weight, and age as covariates in the analysis.
 - Page 2 line 126-130. *“Of these, seven remained significant (choline, TMAO, EPA, methionine sulfate, cer(d18:2/14:0), TG 20:0_34:1 and hex2cer(d18:1/18:0); Fig.1A, Table.S2)) following correction for multiple testing ($q < 0.15$) and adjustment for covariates including enrolment weight-for-age z-score, socioeconomic status, number of months of exclusive breastfeeding, and maternal factors (weight, height, age).”*
 - Page 4 line 164-169 *“Of these nine remained significant when corrected for multiple testing ($q < 0.15$) and adjusted for the covariates enrolment WAZ, number of months of exclusive breastfeeding, maternal age and socioeconomic status (pantothenate, dimethylglycine, lactose, succinate, valine, 2'-FL, NMND, D-galactose, and 3'FL) (Fig.1A; Table.S3). Interestingly the seasonality was dampened in the HMO 2'-FL when adjusted for maternal body composition (weight, height).”*
 - Page 20 line 792-795 *“The analyses were repeated using residualized features according to a linear model including potential covariates (enrolment weight-for-age z-score, socioeconomic status, number of months of exclusive breastfeeding, and maternal factors (weight, height, age)).”*

For the other pressures mentioned by the reviewer,

- Cosinor analysis was performed on the cohort as a whole and stratified by infant sex. Page 4 line 181-185 *“After stratifying by sex, no plasma or urine metabolites exhibited circannual patterns in the females, while two, 17, and six metabolites were noted in the males at six, 12 and 18 months respectively*

after FDR and covariate adjustment (enrolment weight-for-age z-score, socioeconomic status, number of months of exclusive breastfeeding, and maternal factors (weight, height, age)). (Fig.S4; Table.S6).” and displayed in supplementary figure S4. Following this suggestion, we have also added a section in the discussion related to the sex differences to emphasise the importance of this variable.

Fig.S4. The male infant metabolome is more sensitive to environmental pressures. Plasma and urine metabolites that follow significant cosine distributions based upon birth month in males only at A) six months (n = 149), B) 12 months, C) 18 months (n = 141; $q < 0.15$) following adjustment for enrolment weight-for-age z-score, socioeconomic status, number of months of exclusive breastfeeding, and maternal factors (weight, height, age). Colored by birth season of greatest abundance.

- Page 12 line 450-454 “Interestingly, sex differences were also observed where the male metabolome appears more sensitive to seasonal perturbations than the female. This aligns with literature reporting more distinct metabolic differences in adult males following periods of early life undernutrition compared to community controls (40,41), and a greater adverse response of the male fetus to stress during pregnancy and post-natal life (42,43).”
- Regarding weight at sampling, we have performed cosinor analysis on LAZ, WAZ, MAZ (muac-for-age Z-score) and HAZ (head circumference-for-age Z-score) and identify no seasonal effect on these measures. The findings of these analyses have now been included in a table, as well as their reference in the text. As no seasonal effect was observed we did not adjust for this in our models.
 - Page 8, line 306-310. “As with enrolment anthropometry, no significant seasonal waveforms were detected across anthropometric measures (LAZ, WAZ, HCZ [head circumference-for-age Z-scores], and MAZ

[mid-upper arm-for-age circumference (MUAC) Z-scores]] by birth month from three-18 months (Table.S17)."

- Regarding additional measures related to diet, we performed cosinor analysis on food insecurity and identified a seasonal effect. To explore the impact of this on seasonal metabolic variation, cross correlation analysis was performed and indicated a relationship between seasonal metabolic variation and seasonal food insecurity at birth and at the time of sampling.
 - Nevertheless, we acknowledge that additional seasonal pressures remain unaccounted for in our data such as infections and have highlighted these as a limitation in the discussion. See also response to Reviewer 2 comment 11.
3. There is no quality control of the metabolomics data or description of the reliability of the metabolite identification.
- The plasma metabolomes were measured by LC-MS using commercially available targeted metabolomic kits (Biocrates MxP Quant 500). This was performed using standard manufacturers protocols. The kit includes standards for the compounds to ensure reliability of metabolite identification and quantification and incorporates a quality control sample interspersed throughout the analytical run to ensure reliability and reproducibility. This was used for batch correction to minimise technical variation across the different plates. A sentence has been added to the methods section of the manuscript describing this approach to improve clarity.
 - Page 19 line 736-737. *"This kit incorporates internal standards for metabolite identification and quality control samples to assess reproducibility. This data was acquired following the manufacturer's instructions. The complete list of metabolites included is documented in the kit specification (47) and the quantification was achieved using a 7-point standard curve of internal standards"*.
 - Page 19 line 748-750. *"To minimize technical variation across the different plates, batch correction was performed using standard quality control samples included in each plate."*
 - For the NMR analysis, a pooled sample was prepared from all study samples forming a biological QC. This was analysed repeatedly throughout the run to confirm reproducibility. A sentence has been added to the methods to reflect this Page 19 line 716-717: *"A pooled biological QC was prepared and analyzed repeatedly throughout the run to confirm reproducibility."*
4. The authors state that the birth month showed significant association with specific metabolites. However, the results could be potentially explained by the month the samples were collected, and the food-intake/availability at that time. This is directly linked with the birth month, and thus the food intake on the sampling point. There was a short comment that there was a one sampling-month related metabolite. This is not sufficient, and a more comprehensive analysis/description should be added. How was the situation with the urine metabolome? There should be more discussion of this on the manuscript. This is the major issue on this study.
- We agree with the reviewer that this is the major focus of the manuscript.
 - We identified several metabolites that exhibited cosinor variation in urine and plasma samples collected at 6, 12 and 18 months. To identify those that were influenced by the birth month of the infant, we identified metabolites whose peak concentration

remained static in the same infants when sampled six months apart. For example, plasma choline as shown in Figure 1 is highest in infants born November-January at 12 months and remains highest in infants born in November-January sampled 6 months later at 18 months. Therefore, this cannot be “potentially explained by the month the samples were collected” and why we defined these metabolites as birth month related. Similarly, there are metabolites whose peak concentration shifts by 6 months when sampled 6 months later. We define these as sampling month related metabolites. For example, the triglyceride (20:0_34:1) is most abundant in plasma from March-May born infants sampled at 12 months, but most abundant in plasma from September-November infants sampled at 18 months.

- The sampling month and birth month related metabolites are indicated in figure 1A-C with open and closed samples.

Fig.1: Systemic and urinary metabolome variation at 6, 12 and 18 months is influenced by the month of birth. A-C) Bar chart of 6-, 12- and 18-month urinary, and 12- and 18-month plasma metabolites that follow a seasonal waveform based upon month of birth, from cosinor analysis (plasma n = 199; urine 6-month n = 278; 12-month n = 270; 18-month n = 266; $p < 0.05$ & $q < 0.15$ at \geq one sampling point). Color indicates birth season of peak abundance. Solid circle represents birth-month-dependent metabolites (± 1.5 -month difference between birth month of peak abundance at two sampling points), unfilled circle indicates sampling-month-dependent ($>$ four-month difference). D) Mean plasma choline concentration by birth month at 12 and 18 months (bold line; colored by sampling point; n = 199) following cosine distribution (dashed line) with 95% CI overlaid (shaded). E) Median 1H NMR six-month urinary metabolic spectral profile (n = 278). Metabolites identified to significantly fit cosine waveforms based upon birth month are colored by birth season of peak abundance. Urinary metabolite relative abundance obtained by integrating area under spectral regions. F-J) Examples of urinary metabolites at six months that follow a cosine distribution (dashed line) by birth month. Mean relative abundance

with 95% CI overlaid (shaded). EPA, eicosapentaenoic acid; TMAO, trimethylamine N-oxide; 2-PY, N-methyl-2-pyridone-5-carboxamide; NMND, N-methyl nicotinamide; 2'-FL, 2'-fucosyllactose; 3'FL, 3-fucosyllactose; DMG, dimethylglycine.

- Moreover, we explain this process in supplementary figure 1C.

- In the revised manuscript, the contribution of sampling month to metabolic patterns has been enhanced to make the following points:
 - Page 3 line 134-136 “To determine if these metabolic cycles were driven by contemporary pressures at 12 months of life (i.e., time of sampling) or a legacy of the month of birth, plasma was sampled six months later at 18 months”
 - Page 3 line 138-146 “Metabolites whose peak phase remained static between sampling points (± 1.5 -months to account for season) were considered to be influenced by time of birth (Fig.S1). For instance, choline is birth-month-related as November-December born infants persistently had the highest circulating choline at 12 and 18 months, while those born May-June had the lowest (Fig.1C). These patterns were maintained despite variable pressures across these sampling points, including food insecurity, and rainfall. Conversely, those metabolites influenced by sampling time were identified as those with ~six-month (four-eight months) phase difference between the 12-

and 18-month samples and so exhibit seasonal variation within individuals (Fig.S1)."

- In relation to the urinary metabolome Page 4 line 170-171 *"The peak phases of three metabolites, valine, pantothenate and succinate, shifted by ~six months between sampling points and so were considered sampling month dependent."*
 - In figure 1A-C, the circles above the bar chart also denote the metabolites (plasma and urinary) that were observed to follow seasonal patterns across the infant at multiple time points and hence if they were 'birth month related' (no shift) or 'sampling month related' (~6 month shift).
5. Related to urine samples, the results showed several metabolites stated to be linked with breast milk. How many infants were still breastfed at the two later sampling points?
- The breastmilk related metabolites identified in the urinary samples were only observed to follow seasonal variation across the infants in the six-month samples. Mothers were encouraged to breastfeed exclusively for the first six months and as mentioned on page 7 line 247 "81% of infants were exclusively breastfed up to six months of life". Seasonal variation across the infants in the breastmilk related metabolites were not observed in the urine at the latter two sampling points.
6. There is some data given from the breast milk samples, however, the duration of the breastfeeding, and its impact on metabolic profiles is not described and as the sampling time of the breast milk did not fully match with the sampling of blood and urine, there are several possible issues that should be better explored in the study.
- In this cohort, 81% of infants were exclusively breastfed up to six months of life (page 7, line 247). We do not have data on breastfeeding duration after this time point.
 - We aimed to explore the impact of seasonal variation in the breastmilk on the seasonality observed across the metabolomes of the infants. To do this, we performed cross-correlation analysis between the five-month breastmilk samples and the six-month urinary samples. We agree with the reviewer the potential limitations of comparing breastmilk/infant samples at non-matching time points, therefore we only performed such analysis at the timepoints closest in sampling (e.g, 5 – 6 months).
 - Page 8 line 287-293 *"Additionally, to explore transmission of these seasonal patterns to the infant, cross-correlation analysis compared five-month breastmilk vitamins and six-month urinary metabolites. Significant correlations were observed with a range of phase lags (Fig.S8; Table.S15). These included no phase lag between pantothenic acid in the five-month breastmilk and the six-month urine, with the greatest abundance of this vitamin noted in February/March-born infants. This demonstrates the imprinting of seasonal patterns in breastmilk composition on the metabolic signatures of the infant."*
7. Also the BM samples were grouped by birth month while it would be much more logic to group them by season of sample collection.
- We thank the reviewer for this comment, we have updated the manuscript accordingly to improve the readability of the manuscript.

- Page 7 line 252-261 “Cosinor analysis identified four metabolites at one-month, and seven at five months that displayed seasonality based on sampling month, which persist following correction for multiple testing (Fig.3F; Table.S11). Of these, one metabolite exhibited seasonality at both sampling times, with biotin (vitamin B7) consistently highest in milk from mothers who delivered in April (sampled May at one month and September at five months), a period of increasing food availability following the bean harvest in March. Nicotinamide riboside (NR) and nicotinic acid (NAc), precursors involved in the NAD⁺ synthesis pathway, exhibited seasonality in breastmilk. NR at one-month, and NAD⁺ at five months were highest in milk from mothers delivering in April (NR sampled May, NAD⁺ sampled September), whilst NAc was greatest in five-month breastmilk collected from mothers sampled in July.”
- Page 7 line 265-271 “Samples were therefore grouped by season of birth sampling (Sep-Nov, Dec-Feb, Mar-May, Jun-Aug) and variation was assessed by ANOVA or Kruskal-Wallis, and post-hoc tests (FDR adjusted). Mothers sampled between September – November were observed to have the greatest 2'-FL, lacto-N-fucopentaose I (LNFP-1) and glutamine in their milk at one-month postpartum. Additionally, 2'-FL and LNFP-1 were greatest in five-month milk sampled from mothers in June-August (Table.S12).”
- Page 14 line 531-533 “F) Heatmap of 1- and 5-month breastmilk metabolites displaying seasonality based on sampling month from cosinor analysis (n = 292; q < 0.15).”
- Page 20 line 779 “For breastmilk data, cosinor analysis was applied across sampling months.”

8. Maternal diet and food availability is likely to have a major impact of the BM composition.

- We agree that maternal diet and food availability are highly influential on BM composition. To explore this, we chose to perform cross-correlation analysis between the relative abundances of the B vitamins in the breastmilk across the mothers/infants and contemporary food insecurity.
 - Page 7 line 274-281 “We next assessed the impact of the nutritional environment of the mother on the breastmilk metabolome. Cross correlation was performed between seasonal breastmilk B-vitamin-related metabolites and contemporary food insecurity. (Fig.3F; Table.S13). Short phase lags (10-90 days) were noted between peak food insecurity and maximal breastmilk biotin, and NR at one-month and flavin mononucleotide, at five months. Metabolites most abundant in milk sampled during periods of high food availability (110-200 days phase lag from peak food insecurity) included pyridoxamine at one-month; pantothenic acid (vitamin B5), Nac, NAD, pyridoxal-5-phosphate, and biotin, at five months”.
- As the whole breastmilk metabolome and HMOs were only measured in a subset of mothers/infants there was insufficient numbers to perform cross correlation analysis. Therefore, to explore this question we used wilcoxon tests to compare metabolite abundances between mothers who did versus did not report food insecurity at the time of sampling.
 - Page 7 line 282-287 “To assess the impact of food insecurity on the breastmilk metabolome and HMOs measured from the subset of mothers, we

explored biochemical variation in the season-dependent metabolites associated with contemporary maternal food insecurity reports (No food insecurity [n = 40], reported food insecurity [n = 14]; Wilcoxon test $q < 0.05$; Fig. 3G; Table.S14). At one-month, 2'-FL and LNFP-1 were lower in milk from mothers who did not report food insecurity. At five months, only three mothers from this subset reported food insecurity, therefore statistical analysis was not performed."

9. The figure texts lack sufficient description; none of the figures explains what are the panels A, B, C etc.

- We believe this issue relates to the requested format of the journal where the figure and its title is placed at the appropriate location within the manuscript and the full figure legend/description is included at the end. All the requested information is present see page 14-15 line 495-554.

Reviewer #2 (Remarks to the Author):

Overall Comments:

1. Overall, this paper provides valuable insights into birth-season-related effects on infant undernutrition, which can be applied by those planning future interventions in low and middle income countries to maximize intervention effectiveness.
 - We thank the reviewer for recognizing the valuable insights provided by this work.
2. Introduction: In the first paragraph, it would be valuable to briefly mention other potential sources of birth month impacts on infant health and development including pathogen exposure/enteric dysfunction responses to rainfall, before getting into why and how this study chooses to look at the impacts of food insecurity. Though covered in the discussion, these additional sources are important enough to be considered throughout the paper.
 - We have briefly mentioned infection as a modulator for infant health and development in the introductory paragraph and then briefly expanded on this in the second paragraph (see comment 4).
 - Page 2 line 44-45 "Nutritional availability and quality, as well as enteric infections, are major contributors, and in many settings follow seasonal variation."
3. It would also underscore the importance of this study to briefly mention the potential impacts of climate change on seasonal variation in rainfall and food insecurity.
 - A sentence has been added to the introduction highlighting the importance of this study in relation to climate change.
 - Page 2 line 55-57 "As climate change is a major contributor to increasing food insecurity, understanding and preventing its detrimental impact on early life undernutrition remains crucial."

4. Likewise, the second paragraph should include a discussion of the seasonal patterns in pathogen infection in Haydom.
 - *Shigella* has been observed to vary seasonally in this population, with the greatest infection burden during the rainy season. This has been added to the description of the study site in the introduction.
 - Page 2 line 63-65 “*Enteric infections are common in this population and certain pathogens, such as shigella, vary seasonally having the greatest burden in the rainy season.*”
5. In Line 61, it would be helpful to include an overview of the arms in the trial and the impact of the interventions on the primary and/or secondary outcomes of the trial. This would improve the rationale or set up of the third paragraph describing effect measure modification.
 - These details about the ELICIT trial have now been included earlier in the introduction.
 - Page 2 line 67-70 “*These infants were enrolled in the Early Life Interventions for Childhood growth and development In Tanzania (ELICIT) randomized control trial (15-17). This trial had a 2 x 2 factorial design including nutritional nicotinamide and antimicrobial interventions. However, no effect on the primary outcome of 18-month length has been observed.*”
6. Page 2, Lines 87-90: Please provide an overview of the ELICIT trial arms. Include details on the trial set up (i.e., a 2x2 factorial design, administration of antimicrobials (azithromycin and nitazoxanide, randomised together) and nicotinamide). Provide a rationale for why this secondary analysis only used a subset of two arms. Also, provide the total N from these two arms.
 - We have now added the requested information to the manuscript see page 3 line 95-101.
 - “*This is a secondary analysis of the ELICIT cohort (15), following infants born between September 2017-18 longitudinally to 18 months of life. The ELICIT study was a randomized 2 x 2 factorial, double-blind placebo-controlled trial with the aim of improving early life growth using a nicotinamide and/or antimicrobial (azithromycin, nitazoxanide) intervention. Here, data from the placebo group (n = 278) and infants receiving nicotinamide (n = 276, breastfeeding mothers provided 250 mg daily tablets for six months post-delivery, followed by direct infant supplementation (100 mg sachets) from six to 18 months) was studied.*”
 - There are several reasons why the antimicrobial group was not included in this analysis. Firstly, the antimicrobials were only provided as a single dose at 6, 9, 12, and 15 months and the urine and plasma samples were collected prior to administration of the antimicrobials. As such, from a metabolic perspective, the enduring effects of an antimicrobial are unlikely to be reflected in the metabolome. However, it is not possible to rule out carry over of the treatment. Secondly, 54–86% of children were observed to receive some non-study antibiotic during the month prior to the 6-, 12-, and 18-month biological sample collection. Therefore, this limits the ability to draw conclusions from a comparison between control versus antimicrobials.

Lastly, the antimicrobial was not observed to induce any improvements in development. We have now included an explanation in the main manuscript text and the methods section, as well as N for these two arms.

- Page 3 line 106 “The antimicrobial group was not considered for this analysis as the biological samples were collected prior to the administration of a single-dose of the antimicrobials at 6, 12, and 18 months and their long-term biochemical consequences cannot be excluded. Additionally, receipt of non-study antibiotics was common (54 - 86% infants).”
 - Page 16 line 601-603 “This divided the individuals to receive either nicotinamide plus placebo ($n = 276$), antimicrobial plus placebo ($n = 277$), both nicotinamide and antimicrobial ($n = 267$), or both placebos ($n = 278$). In this manuscript, only infants from the placebo/placebo and arms nicotinamide/placebo arms of the study were considered. The seasonality analysis in this manuscript included placebo/placebo arm infants to observe seasonal patterns unaffected by the intervention. Only infants from the nicotinamide/placebo arm are included when considering the intervention to separate the individual intervention effects. The antimicrobial intervention was not considered in this manuscript as the biological samples were collected prior to the administration of the single-dose antimicrobials. Additionally, receipt of non-study antibiotics was common (54 - 86% infants).”
7. Page 6, Lines 204-205: It is unclear what this hypothesized relationship is based on - perhaps missing reference or reference to supplementary data - “These were all highest in infants born during peak food insecurity (November-February) and may be associated with maternal-infant wasting around the time of delivery.”
- To clarify, the outlined metabolites measured at 12 and 18 months are related to the month in which the infant is born. Those with the highest plasma concentrations were born during peak food insecurity (preharvest). During peak food insecurity wasting is expected to be most prevalent. However, we have omitted this statement.
8. Figure 3: Panel labels effective here; would be nice to see these added to other figures. For Panel F though, the subtitle feels misleading since numbers represent phase lag in days rather than any seasonal variation – consider changing the label to “Peak phase season variation in breastmilk vitamins”
- We thank the reviewer for this suggestion and have made the requested change to figure 3. We have also added titles to the other figures. For example, in Figure 1 and Figure 2 shown below.
 - Figure 3:

Fig.3: Annual variation in environmental factors across pre- and postnatal periods contribute to metabolic seasonality. A) Mean monthly (shaded), and 8-year average rainfall (dashed line) in Haydom, Tanzania. B) Monthly food insecurity (% mothers worried about food availability; black line; 95% CI shaded) reported over 18-month period following enrolment. Annual patterns of rain and harvests highlighted. C-E) Lollipop plots displaying phase lags between peak food insecurity during birth year (January 2018) and the birth month of greatest plasma and urinary metabolite abundance at 6 (n = 278), 12 (n = 270) and 18 months (n = 266; q < 0.05) from cross-correlation. Color indicates phase lag; shape indicates source, and size R². Shaded area highlights birth-month-dependent metabolites. F) Heatmap of 1- and 5-month breastmilk metabolites displaying seasonality based on sampling month from cosinor analysis (n = 292; q < 0.15). Colors indicate highest abundance sampling season. Values indicate phase lags (days) between peak food insecurity and peak metabolite abundance from cross-correlation (q < 0.05). G) Bar chart visualizing log₂ fold change of 1-month breastmilk metabolite expression by reported food insecurity at sampling (No n = 40; yes n = 14; FDR p < 0.05, Wilcoxon test). FAD, flavin adenine dinucleotide; FMN, flavin mononucleotide; PLP, pyridoxal phosphate; Nac; nicotinic acid; Pa, pantothenic acid; NAD, nicotinamide adenine dinucleotide; NR, nicotinamide riboside; PM, pyridoxamine; PN, pyridoxine; 2'FL, 2-fucosyllactose; LNFP-1, lacto-N-fucopentaose I; GABA, γ -aminobutyric acid; 3'FL, 3-fucosyllactose; NMND, N-methylnicotinamide; EPA, eicosapentaenoic acid; C18.2, octadecadienylcarnitine.

Fig.1: Systemic and urinary metabolome variation at 6, 12 and 18 months is influenced by the month of birth. A-C) Bar chart of 6-, 12- and 18-month urinary, and 12- and 18-month plasma metabolites that follow a seasonal waveform based upon month of birth, from cosinor analysis (plasma $n = 199$; urine 6-month $n = 278$; 12-month $n = 270$; 18-month $n = 266$; $p < 0.05$ & $q < 0.15$ at \geq one sampling point). Color indicates birth season of peak abundance. Solid circle represents birth-month-dependent metabolites (± 1.5 -month difference between birth month of peak abundance at two sampling points), unfilled circle indicates sampling-month-dependent ($>$ four-month difference). D) Mean plasma choline concentration by birth month at 12 and 18 months (bold line; colored by sampling point; $n = 199$) following cosine distribution (dashed line) with 95% CI overlaid (shaded). E) Median 1H NMR six-month urinary metabolic spectral profile ($n = 278$). Metabolites identified to significantly fit cosine waveforms based upon birth month are colored by birth season of peak abundance. Urinary metabolite relative abundance obtained by integrating area under spectral regions. F-J) Examples of urinary metabolites at six months that follow a cosine distribution (dashed line) by birth month. Mean relative abundance with 95% CI overlaid (shaded). EPA, eicosapentaenoic acid; TMAO, trimethylamine N-oxide; 2-PY, N-methyl-2-pyridone-5-carboxamide; NMND, N-methyl nicotinamide; 2'-FL, 2'-fucosyllactose; 3'FL, 3-fucosyllactose; DMG, dimethylglycine.

Fig.2: Influence of birth seasonality on cognitive outcomes and related metabolites. A) Mean language MDAT (Malawi Developmental Assessment Tool) score by birth month at 18 months (bold line; $n = 199$) following cosine distribution (dashed line; $p = 0.0006$) with 95% CI overlaid (shaded). B) Lollipop plot visualizing phase lags (months) between birth month of greatest 18-month language MDAT scores and greatest seasonal 12-month plasma metabolites ($n = 199$; FDR $p < 0.05$) from cross-correlation analysis. Size indicates R^2 . C-D) Mean concentration of plasma trimethylamine N-oxide (TMAO) and eicosapentaenoic acid (EPA) by birth month following cosine distribution (dashed line) at 12 and 18 months (bold line; colored by time point; $n = 199$) with 95% CI overlaid (shaded).

9. Figure 4: Similarity in color between upper and lower tertile and the other key (control and nicotinamide) makes it difficult to distinguish them. Would also suggest moving control and nicotinamide key somewhere more central so it resembles the other legends or making it larger/more obvious. Would recommend rearranging panels so that axes and deeper similarities align more. Ex: Panels A,G,H would be easier to understand in a row; Panels B/C, D/E would make more sense stacked atop one another.
 - We thank the reviewer for this suggestion and have changed the colours for clarity. We have also moved the figure legend to be more visible. However, due to the order the data is discussed in the results, we cannot reorder the figures.

Fig.4: Nicotinamide availability influences infant growth and can be improved through maternal and infant supplementation. A) Differences in 0–18-month LAZ (length-for-age Z-score) of non-supplemented infants persistently in top ($n = 22$) and bottom ($n = 16$) urinary N-methyl nicotinamide (NMND) tertiles (6, 12 and 18 months) assessed by time series analysis. Mean relative abundance of B-C) 1-month breastmilk nicotinamide riboside (NR) and D-E) 6-month urinary NMND following cosine (dashed line) with 95% CI (shaded) in control infants (NR $n = 292$, NMND $n = 278$) and nicotinamide supplemented (NR $n = 295$, NMND $n = 276$). Colored by group. F) Correlation between enrolment WAZ (weight-for-age Z-score; WAZE) and 12-month LAZ (LAZ₁₂) for control and nicotinamide supplemented infants. Colored by birth season. Significance from Fisher's Z-transformation and Z test of correlation coefficients (Nicotinamide Dec-Feb born $n = 114$, control $n = 116$; nicotinamide Jun-Aug born $n = 55$, control $n = 55$). G-H) Differences in 0–18-month LAZ for children born with top tertile enrolment WAZ ($WAZ \geq -0.13$) in Jun-Aug (nicotinamide $n = 21$, control $n = 18$) and Dec-Feb (nicotinamide $n = 30$, control $n = 21$) assessed by time series analysis. Colored by group.

10. Discussion: A thorough discussion of birth month effect modification by different factors (by sex, SES, enrolment WAZ etc. as mentioned in results pg 4 line 150-153) is warranted here. Since these differential effects have major implications for intervention planning, they are important to interpret and discuss.

- We have now added the following paragraph to the discussion section expanding the discussion on birth month effect modifiers.
- Page 12, line 443-452.
 - *“It should be noted that other birth month effect modifiers on the metabolome are likely present. This includes factors that directly impart the seasonal metabolic associations such as infections, and those that shape the*

magnitude of seasonal variation such as socioeconomic status. For example, seasonal variation in cognitive outcomes and their association with plasma metabolites were only seen in those with the highest socioeconomic status (top 50%). Interestingly, sex differences were also observed where these findings indicate the male metabolome is more sensitive to seasonal perturbations than the female. This aligns with literature reporting more distinct metabolic differences in adult males following periods of early life undernutrition compared to community controls (40,41), and a greater adverse response of the male fetus to stress during pregnancy and post-natal life (42,43).”

11. Page 12, Lines 433-435: This could be expanded to include more specifics, especially regarding effects of rainfall on enteropathogen exposure and potentially EED. This underscores the importance of targeted interventions as children in the crucial developmental phase that encompasses the first six months of life may be facing the dual challenges of increasing food insecurity and increased exposure to environmental pathogens during the rainy season.

- We thank the reviewer for this interesting point. As we do not have data on enteropathogens we are restricted in the specifics we can discuss, we have now added sentences to the introduction and limitations section of the discussion highlighting this as an area of importance and future research.
 - Page 12 line 470-474 *“For example, variation in gut microbial composition, infection burden, immune responses and enteric dysfunction may also change depending on the time of year an infant is born and could contribute to these metabolomic differences (44-46). Enteropathogenic infections are prevalent in this setting (47), with certain infections exhibiting seasonal variation, such as increasing Shigella during the rainy season (48).”*
- Additionally, enteropathogenic exposure in this population has been discussed in detail in other manuscripts such as DOI: 10.1371/journal.pone.0294110 which have now been referenced in this work.

Minor Comments:

10. Page 1, Line 27-28: “contributions of this to these adverse phenotypic outcomes” feels awkward due to unnecessary repetition of “this”. Consider changing to “it is unclear if this imprints on the developing metabolome and contributes to adverse phenotypic outcomes”

- We have updated this in the manuscript.

11. Page 1, Line 30: consider changing “persisting up to 18 months of life” to “persisting up to at least 18 months of life” to clarify that this finding is due to length of study.

- We have updated this in the manuscript.

12. Page 2, Line 48: consider deleting “its influence”.

- We have updated this in the manuscript.

13. Page 2, Line 72-73: should reword “the tryptophan-niacin-NAD⁺ pathway is perturbed with early life undernutrition and is related to growth” to “the tryptophan-niacin-NAD⁺ pathway is related to growth and perturbed with early life undernutrition” to simplify and clarify importance.

- We have updated this in the manuscript.

14. Page 13, Lines 474-488: Figure 3 caption missing extensive list of acronyms and abbreviations.

- These have now been added. **“Fig.3: Annual variation in environmental factors across pre- and postnatal periods contribute to metabolic seasonality.** A) Mean monthly (shaded), and 8-year average rainfall (dashed line) in Haydom, Tanzania. B) Monthly food insecurity (% mothers worried about food availability; black line; 95% CI shaded) reported over 18-month period following enrolment. Annual patterns of rain and harvests highlighted. C-E) Lollipop plots displaying phase lags between peak food insecurity during birth year (January 2018) and the birth month of greatest plasma and urinary metabolite abundance at 6 (n = 278), 12 (n = 270) and 18 months (n = 266; FDR p < 0.05) from cross-correlation. Color indicates phase lag; shape indicates source, and size R². Shaded area highlights birth-month-dependent metabolites. F) Heatmap of 1- and 5-month breastmilk metabolites displaying seasonality based on delivery month from cosinor analysis (n = 292; q < 0.15). Colors indicate highest abundance delivery season. Values indicate phase lags (days) between peak food insecurity and peak metabolite abundance from cross-correlation (q < 0.05). G) Bar chart visualizing log₂ fold change of 1-month breastmilk metabolite expression by reported food insecurity at sampling (No n = 40; yes n = 14; q < 0.05, Wilcoxon test). FAD, flavin adenine dinucleotide; FMN, flavin mononucleotide; PLP, pyridoxal phosphate; Nac, nicotinic acid; Pa, pantothenic acid; NAD, nicotinamide adenine dinucleotide; NR, nicotinamide riboside; PM, pyridoxamine; 2’FL, 2-fucosyllactose; LNFP-1, lacto-N-fucopentaose I; GABA, γ-aminobutyric acid; 3’FL, 3-fucosyllactose; 2-PY, N-methyl-2-pyridone-5-carboxamide; NMND, N-methylnicotinamide; EPA, eicosapentaenoic acid.”

Reviewer #3 (Remarks to the Author):

1. This is an interesting and well-conducted study, showing that seasonal availability of food affects the infant metabolome in a low-income context, with potential implications for health and development.
 - We thank the reviewer for their complementary words regarding our manuscript.
2. The study does not state any clear hypotheses and there are no sample size calculations. Could the authors clarify what aspects of the analysis plan were defined a priori, including e.g. decisions on seasonal bins, length of phase lag and stratification analyses? The obvious concern here is the risk of 'false positives', given the multiple outcomes from metabolomics conducted on plasma and urine samples. The authors attempt to address this using false discovery rate corrections. I am not qualified to comment in detail on the statistics, but do request the authors justify the decision to apply FDR corrections at $p < 0.25$ for some analyses as reported at lines 454 and 484, rather than the 'typical' $p < 0.05$.
 - The ELICIT study was planned as an RCT to assess whether nicotinamide and antimicrobials improve growth outcomes in infants from Haydom, Tanzania, a resource limited setting.
 - The metabolomic component was included to assess the biochemical changes induced by these interventions and their associations with growth. The primary outcome of this study was improvement in LAZ at 18-months and was not significant. This work represents a secondary, exploratory analysis, exploring seasonal variation in the metabolome and factors driving it and potential relationships with developmental outcomes. As is standard with metabolomic approaches, to minimise risk of false positives, steps are taken to adjust for multiple testing. Correcting for multiple testing is especially difficult however in metabolomics analysis because of the wealth of metabolites measured, which are not truly independent and can covary due to their shared metabolic pathways, functions and complex network of interactions. As such, conservative threshold values can result in a high number of false negatives. For the cosinor analysis, >600 metabolites were measured by mass spectrometry and NMR spectroscopy and so an FDR corrected threshold of 0.25 was used. This identified seasonal candidate metabolites which were subsequently used for cross correlation analysis. A stricter FDR threshold was employed for correlating these metabolites with additional variables (e.g., food insecurity, rainfall, cognition data) as the same covariation issue is not encountered.
 - As several metabolites exhibit seasonal patterns at multiple time points, we believe this demonstrates the biological relevance and validity of our findings. At a stricter FDR, some of these findings are lost (e.g PC ae 42:2 following adjustment for enrolment WAZ and number of months of exclusive breastfeeding would be significant at 18 months ($p = 0.00006$; $q = 0.017$) but not at 12 months ($p = 0.006$; $q = 0.20$). We therefore believe this would be introducing a greater number of false negatives and potentially masking biologically important results.
 - However, in response to this suggestion, we have increased the FDR-corrected threshold to 0.15 to reduce the risk of false positives. When assessing patterns across multiple time points, we have included those with a q-value < 0.15 for at least one time point, as this criterion strengthens confidence in their presence at other time points by lowering the probability of spurious findings. An FDR threshold of 0.15 has been employed in many metabolomics studies previously, e.g.

- DOI: 10.1038/s41598-018-35372-w
 - DOI: 10.1002/ijc.32218
 - DOI: 10.1038/s41398-022-01856-7
 - DOI: 10.3390/ijms24119736.
 - The analysis has been repeated with this stricter FDR threshold and the figures, tables and text have been updated accordingly throughout the manuscript.
 - Page 3 line 126-130 *“Of these, seven remained significant (choline, TMAO, EPA, methionine sulfate, cer(d18:2/14:0), TG 20:0_34:1 and hex2cer(d18:1/18:0); Fig.1B)) following correction for multiple testing ($q < 0.15$) and adjustment for covariates including enrolment weight-for-age z-score, socioeconomic status, number of months of exclusive breastfeeding, and maternal factors (weight, height, age).”*
 - Page 3 line 136 *“. Several plasma metabolites exhibited circannual patterns across the infants at 18 months ($n = 27$; $n = 6$ following FDR correction ($q < 0.15$) and adjustment for confounders; Fig.1C; Table.S2).”*
 - Page 4 line 164-167 *“Of these nine remained significant when corrected for multiple testing ($q < 0.15$) and adjusted for the covariates enrolment WAZ, number of months of exclusive breastfeeding, maternal age and socioeconomic status (pantothenate, dimethylglycine, lactose, succinate, valine, 2'-FL, NMND, D-galactose, and 3'FL) (Fig.1A; Table.S3).”*
 - Page 4 line 181-185 *“After stratifying by sex, no plasma or urine metabolites exhibited circannual patterns in the females, while two, 17, and six metabolites were noted in the males at six, 12 and 18 months respectively after FDR and covariate adjustment (enrolment weight-for-age z-score, socioeconomic status, number of months of exclusive breastfeeding, and maternal factors (weight, height, age)) (Fig.S4; Table.S6).”*
 - Page 14 line 497-501 *“Fig. 1: Systemic and urinary metabolome variation at 6, 12 and 18 months is influenced by the month of birth. A-C) Bar chart of 6-, 12- and 18-month urinary, and 12- and 18-month plasma metabolites that follow a seasonal waveform based upon month of birth, from cosinor analysis (plasma $n = 199$; urine 6-month $n = 278$; 12-month $n = 270$; 18-month $n = 266$; $p < 0.05$ & $q < 0.15$ at \geq one sampling point).”*
 - Page 14 line 531-533 *“F) Heatmap of 1- and 5-month breastmilk metabolites displaying seasonality based on sampling month from cosinor analysis ($n = 292$; $q < 0.15$).”*
 - Page 20 line 785-789 *“All p-values were corrected for multiplicity of testing according to Benjamini-Hochberg. Features with a $q < 0.15$ for at least one sampling time point were considered significant cosine fits and included in further analysis. This threshold was selected due to the large number of metabolites measured and the covariation that exists across the metabolomic dataset through shared metabolic pathways or functions.”*
3. The decision to look only at the placebo/placebo arm and nicotinamide/placebo arms of the study (line 543) requires justification.
- Please see response to reviewer 2, comment 6.
4. Full details on sample collection and analysis would be good, bearing in mind that sample handling is so important for metabolomics. The level of detail is insufficient for reproducibility.

- We have now added more details to the methods section regarding sample collection and transfer.
 - Page 18 line 677-689 *“Blood samples were collected at 12 (placebo n = 199; nicotinamide n = 199), and 18 (placebo n = 199; nicotinamide n = 198) months of life via phlebotomy, transported on ice to the laboratory to be processed into plasma, and stored at -80°C for shipping to the United Kingdom for metabolomic analysis. Urine samples were collected into a sterile bag at six (placebo n = 278; nicotinamide n = 276), 12 (placebo n = 270; nicotinamide n = 267), and 18 months (placebo n = 266; nicotinamide n = 263), before being transported on ice to the research site and then stored at -80°C to be shipped to the United Kingdom for metabolomic analysis. Breastmilk was collected at one- and five-months post-partum (placebo n = 292; nicotinamide n = 295) at the midpoint of a feeding session. The area around the areola was cleaned using soap and water and rinsed with deionized water prior to sampling. Approximately 8 mL was expressed by hand into a sterile container, which was placed on ice for transportation to the laboratory. Here, the samples were aliquoted whilst being shielded from light and stored at -80°C and shipped to the United States of America for metabolomic analysis.”*

- 5. The authors should discuss why seasonal relationships in plasma and urine metabolite concentrations are seen only in males but not in females (line 151).

- We have added a section on this in the discussion.
 - Page 12 line 448-452. *“Interestingly, sex differences were also observed where the male metabolome appears more sensitive to seasonal perturbations than the female. This aligns with literature reporting more distinct metabolic differences in adult males following periods of early life undernutrition compared to community controls (40,41), and a greater adverse response of the male fetus to stress during pregnancy and post-natal life (42,43).”*

- 6. Looking at the seasonal patterns in metabolite concentrations (Figure 1) – the charts are plotted by month, from September to August. For several metabolites including choline, the concentrations in September and August are substantially different with no overlap in the 95% CI, yet these are adjacent months. How can this be explained? To me, this points towards other, non-seasonal drivers, possibly in combination with large sampling or measurement error when sample sizes are subset by month?

- As shown in Fig3A-B there was much greater food insecurity in the first year of the study compared to the second, following the drought demonstrated by the much lower than average rainfall. Therefore, differences are to be expected between the environmental conditions of September 2017 and September 2018 that the infants were experiencing. To account for this, the cosine analysis was performed including a sloping mesor, to allow for year-on-year variation to be incorporated into the model. We believe that this demonstrates the importance of performing such analysis in areas experiencing extreme climatic conditions to understand the consequences such events are having on infant health and development.

- 7. The study assumes a pathway from season to food insecurity to infant metabolome. There are, however, other potential pathways between season and infant

metabolome that merit consideration. For example, moisture and temperature can affect cereal grain metabolism post-harvest. The aspartate data stand out to me, given associations between climate conditions during the growing of cereals and the free asparagine concentration in the resultant crop, e.g. Curtis et al., *Journal of Agricultural and Food Chemistry* 2009 57:1013-1021 DOI: 10.1021/jf8031292. Also, there may be seasonal variation in the geographic source of food items, e.g. more home-produced food in months after harvest, and more purchased food (likely grown further from the home) in the 'lean' season. Asparagine (and potentially other) amino acid concentrations, and other chemical entities including mineral micronutrients, are likely to vary depending on the location of production, including due to soil factors (e.g. Gashu et al., *Nature*. 2021;594:71-6), hence there will be seasonal variability in nutrient intakes independent of experience of food insecurity, and potentially important spatial variability in associations between season of birth, nutrient intakes and the infant metabolome. At line 436, the authors suggest that birth month acts as a proxy for these combination of seasonal drivers – however, these diverse pathways could lead to very different relationships between season and metabolite concentrations in other geographic settings.

- We thank this reviewer for this suggestion and is an area of great interest to us which we are hoping to explore in a future experiment. Given our current data we are restricted in the amount of inference we can draw on this. However, we have added this point to the discussion and raised this as an area for future research in our limitations.
 - Page 12 line 475-480 *“Seasonal fluctuations in post-harvest storage conditions, particularly variations in temperature and moisture (47), can alter the metabolic profiles of crops over time, as well as location of production (48), potentially affecting nutrient and amino acid levels independently of food insecurity. This may alter the relationships between season and metabolite concentrations in other geographic settings.”*
- 8. The authors examined seasonal associations between metabolites and cognitive outcomes. At line 377, the authors suggest the observed associations demonstrate developmental relevance of metabolic seasonality. I think this statement is not sufficiently evidenced by the findings of this exploratory analysis. There are many pathways between season of birth and cognitive development, and many potential confounders for the association between metabolite concentrations and cognitive test scores. For example, labour patterns are likely to be seasonal which may affect breastfeeding patterns, and evidence from other settings including the UK shows associations between season of birth and cognitive development outcomes with hypothesised pathways including seasonal variability in maternal vitamin D status and opportunities for social interactions. And, of course, seasonal variability in consumption of food groups including fruits and vegetables could influence the infant metabolome and cognitive development without a direct interaction between the latter two. The authors do note limitations at line 431, but the inherent limitations of the study design are given insufficient weight and I suggest the authors use more cautious language in suggesting a direct relationship between metabolites and cognitive outcomes.
- We thank the reviewer for these explanations and have dampened down our language regarding the developmental links accordingly.

- Page 10 line 403-405 *“The seasonality observed in the 18-month cognitive outcomes demonstrates developmental relevance of this metabolic seasonality. Interestingly, seasonal variation was also observed in 18-month cognitive outcomes, where infants born in January-February had the greatest MDAT scores at 18-months.”*
9. According to Figure 3A, reported food insecurity in the lean season was greater in 2017 than 2018 and probably 2019. Was 2017 a particularly bad year? The rainfall data suggest two false starts to the rains and a late, intense and short rainy period. Could the authors comment on this, and any potential implications for the wider external validity of the seasonal associations observed?
- Yes, there was a drought during year prior to recruitment which also associated with greater food insecurity during the year of recruitment. We acknowledge the possibility that the extreme weather conditions intensified the seasonal metabolic variation observed. However, as Reviewer 2 (Q3) commented about the implications of climate change, with increasing extreme weather conditions such as droughts and rainfall, understanding the consequences of such events on child development is a key priority. If the metabolic seasonal variation was enhanced by the extreme weather conditions and resultant food insecurity, this emphasises the importance of this work for other settings experiencing food scarcity. We therefore hope this work will encourage further research into this important area.
10. I am not convinced that you would deliver any common public health interventions differently on the basis of this study, when it is already well-known that the hungry season brings food security and nutrition-related challenges. Perhaps the authors can explain which aspects of common interventions or which particular interventions might be adapted or tailored for seasonality, where this does not already occur? To my knowledge, supplementation with choline or nicotinamide are not common public health interventions. However, the study could be used to inform the design of relevant future studies including monitoring and evaluation studies, including in terms of sampling timings.
- The aim of this work was to highlight the potential for seasonal metabolic variation and demonstrate that it is an important consideration when designing interventions in such settings. This work provides evidence to justify its consideration for future studies and insights to support the design and implementation of future seasonal interventions.
11. The inclusion and ethics section lacks detail. Please could the authors provide information on the ethical review committees and relevant application reference numbers? Also, please clarify whether the current study, i.e. analysis of infant metabolites, was included in the main ELICIT study protocol as reviewed and approved by the relevant ethics committees, or whether the current study operated under separate protocols and approvals? This section should also specify information on data and material transfer agreements.
- Information on the ethical review committees and application numbers have now been added to this section of the methods. Yes, the metabolomic analysis was included in the main ELICIT study protocol.
 - Page 14 line 574-580 *“The study, and transfer of data and materials, were approved and was closely followed by all local research regulatory agencies,*

including the Tanzanian National Institute for Medical Research (NIMR, reference number HQ/R.8a/Vol.IX/2424), the University of Virginia Health Sciences Research Institutional Review Board (HSR-IRB, reference #19465), and Tanzanian Food and Drug Administration (reference number TFDA0017/CTR/0005/02). The study also received ethical approval from the University of Southampton Research Ethics Committee (reference number 61337.A1).”

Minor comments

12. Line 214. This should be seasonal variation in, not via, breastmilk

- We have updated this in the manuscript.

13. Figure S1. There is a typo in section C, i.e. ‘individuals’

- We have updated this in the manuscript.

Response to reviewers

We are delighted that Reviewers 2 and 3 appreciated our comprehensive responses to their remarks and are now satisfied with the manuscript. We have made the modest changes requested to the manuscript (latest modifications highlighted in the manuscript in green; responses to first review highlighted in red).

Reviewer #2:

Thank you for comprehensively addressing my comments. A few additional comments below.

Minor Comments:

Line 98-100: “Here, data from the placebo group (n = 278) and infants receiving nicotinamide (n = 276, breastfeeding mothers provided 250 mg daily tablets for six months post-delivery, followed by direct infant supplementation (100 mg sachets) from six to 18 months) was studied.”

The layered parenthetical is obscure and difficult to understand. The additional details about how the nicotinamide was introduced are important and deserve their own sentence.

This has been amended to read:

Page 3, Line 98-100: “Nicotinamide was provided to breastfeeding mothers (250 mg daily tablets for six months post-delivery) followed by direct infant supplementation (100 mg sachets) from six to 18 months. Here, data from the placebo group (n = 278) and infants receiving nicotinamide (n = 276) was studied.”

Line 56-57: “As climate change is a major contributor to increasing food insecurity, understanding and preventing its detrimental impact on early life undernutrition remains crucial.”

This sentence would benefit from a more thorough transition, clarifying how seasonality is related to anticipated changes in climate, especially regarding droughts.

We have now added the following sentence to clarify this point.

Page 2, Line 56-59: “Climate change has been shown to exacerbate seasonal variations in environmental conditions and is anticipated to increase the frequency and severity of droughts and food insecurity (13). As such, understanding the detrimental impact of adverse climatic conditions on early life undernutrition and its impact on development remains crucial.”

Reviewer #3:

Thank you for considering and addressing my comments. I am happy with the changes made.

Reviewer #1:

In contrast to the other reviewers, Reviewer 1 still had queries about the work and our responses to earlier comments. Many of these can be answered by information already contained within the manuscript and its supplementary information. We have addressed the comments for Reviewer 1 below and signposted information within the manuscript that should clarify areas of misunderstanding.

While the manuscript has improved, there are now several parts on the new data that is making the interpretation more confusing, and it is still not fully clear that the birth month is the driving factor is many of the observations listed here. The main concerns are that of the seven metabolites that after adjustment are linked with birth month (choline, TMAO, EPA, methionine sulfate, cer(d18:2/14:0), TG 20:0_34:1 and hex2cer(d18:1/18:0) most of them are also linked with seasonal variation (choline, TMAO, EPA, cer(d18:2/14:0), TG 20:0_34:1 and hex2cer(d18:1/18:0). This is not clearly discussed in the manuscript. Which one is the main driving force, the birth month or the season?

- This is the major focus of the manuscript, and our approach has been designed to specifically answer this question (*i.e.* distinguish between birth month and sampling month-related metabolites).
- We previously included a schematic to explain the approach in the supplementary information (Supplementary Figure S1).
- This is clearly explained in the manuscript and in our 'Response to Reviewers' document.

In addition, the associations seem to be rather weak, as FDR correction of 0.15 was used, normally one would use FDR= 0.05 or in some case, FDR of 0.1.

- We disagree with the reviewer on this point. The metabolites being discussed are those whose plasma/urine abundance fits a cosine wave across >200 infants depending on the month in which they were born and/or month they were sampled.
- An FDR correction (0.15) was then applied to the significance of these cosine fits (across >500 molecules), further reducing the risk of false positives.
- Several metabolites exhibit seasonal patterns at multiple time points, reinforcing the biological relevance and validity of our findings.
- As such, this is not a weak association.
- We are happy to see that Reviewer 3 agrees with this approach.
- Correcting for multiple testing is especially difficult in metabolomics analysis because of the wealth of metabolites measured. Many of these molecules are not truly independent and can covary due to their shared metabolic pathways, functions and complex network of interactions. As such, conservative threshold

values (FDR = 0.05) are overly strict and can result in a high number of false negatives.

- The FDR threshold of 0.15 is a good compromise to avoid false positives while limiting the amount of false negatives. This has been employed in many metabolomics studies previously, e.g.
 - DOI: 10.1038/s41598-018-35372-w
 - DOI: 10.1002/ijc.32218
 - DOI: 10.1038/s41398-022-01856-7
 - DOI: 10.3390/ijms24119736.

How would the results look if more stringent FDR would be used?

- As expected, at a stricter FDR, some of these findings are lost. For example, TG (20:0_34:1) following adjustment for enrolment WAZ, number of months of exclusive breastfeeding, socioeconomic status and maternal factors is significant at 18 months ($p = 0.0005$; $q = 0.06$) at a stricter FDR, but not at 12 months ($p = 0.001$; $q = 0.11$). Using a more stringent FDR would introduce a greater number of false negatives and potentially mask biologically important results.
- Supplementary Tables 2 and 3 shows the results for the plasma and urinary metabolomes with their q values so that readers have all the information requested by the reviewer.
- From here, 28 plasma metabolites are significant at 0.25, 11 at 0.15, 7 at 0.10 and 5 at 0.05. Following adjustment for covariates, 23 plasma metabolites are significant at 0.25, 10 at 0.15, 7 at 0.1 and 5 at 0.05. Additionally, 17 urinary metabolites are significant at 0.25, 17 at 0.15, 15 at 0.1, and nine at 0.05. After covariate adjustment, 14 urinary metabolites are significant at 0.25, nine at 0.15, six at 0.1 and four at 0.05.

Overall, it would be good to have a similar plot that is shown in Fig1D for choline for all metabolites that were listed significant, showing also the error range of measurements, both related to the birth month and for month of sampling.

- This has been shown in Figure 1E-J for several urinary metabolites.
- We have added these plots for the other plasma metabolites in Supplementary Figure S3 and urinary metabolites in Supplementary Figure S4.

Fig.S3.

Variation in plasma metabolome at 12 and 18 months is influenced by the month of birth

A-L) Mean plasma metabolite concentrations by birth month at 12 (grey) and 18 (green) months (bold line; $n = 199$) following cosine distribution (dashed line) with 95% CI overlaid (shaded). Metabolites plotted at time point observed to follow a seasonal waveform based upon month of birth, from cosinor analysis (plasma $n = 199$; $p < 0.05$ & $q < 0.15$ at \geq one sampling point).

Fig.S4.

Variation in urinary metabolome at 6, 12 and 18 months is influenced by the month of birth
 A-P) Mean urinary metabolite relative abundances by birth month at six (pink), 12 (grey) and 18 (green) months (bold line) following cosine distribution (dashed line) with 95% CI overlaid (shaded). Metabolites plotted at time point observed to follow a seasonal waveform based upon month of birth, from cosinor analysis (6-month $n = 278$; 12-month $n = 270$; 18-month $n = 266$; $p < 0.05$ & $q < 0.15$ at \geq one sampling point).

The circannual patterns were adjusted with sex, and after that, no significant patterns were observed in female infants. Why was not the birth month related analyses stratified by sex?

- This was performed (see Fig S4).
- We have added the following sentence to the main text:

Page 4, line 188-190: “Here, plasma choline and TMAO and urinary D-galactose were noted to be birth month dependent metabolites in males while urinary pantothenate and lactose were sampling month dependent”.

Urine metabolomics, why only the impact of birth months has been investigated, and the impact of sampling month has not been done in a similar manner than was done for plasma metabolomics?

- The same analysis was performed in the urine and plasma samples which is clearly displayed in Figure 1A-C and explained in the manuscript.

PCA model shown in Fig S3, is the sampling month considered in the model? How the model would look if the grouping would be done according to the sampling month, rather than birth month?

- This will not alter the model.
- Here, the PCA model was built using metabolites measured at one time point (e.g. 6, 12, or 18 months) as the descriptor matrix. PCA is an unsupervised approach based on variance and does not consider birth month or sampling month.
- In the resulting scores plot from the PCA model (shown in FigS3), the observations (infants) are coloured according to the infant's birth month (e.g. September-November vs March-May).
- If the observations are coloured according to sampling month, the scores plot will look the same because the September-November born infants will have one colour (as sampled March-May) while March-May born infants will have a different colour as sampled (September-November).

Figure 1 E-J are not explained in the text. Fig 1 E is not very informative.

- For clarity we have directed the reader to each individual panel in Fig1 when the relevant urinary result is stated in the manuscript text (see page 4, lines 164-177).
- However, we disagree with the comment that Fig 1E is not very informative. This is an ^1H NMR spectrum where peaks that follow a circannual rhythm are highlighted and coloured by the birth month at which they are highest in the urine. This captures a large amount of information in a single figure.

Line 109-110: The authors have added information that large part of the cohort has been getting antibiotics. Use of antibiotics can have a significant impact on metabolome, was this considered in the analyses? Is the use of antibiotics related to season?

- We have limited information regarding any exposure to antibiotics (Yes, No) in the time prior to sampling. This included up to 86% of the population and intake did not differ by birth month.
- We have added this information to the main text: Page 3, lines 114-115: "Additionally, receipt of non-study antibiotics was common (54 - 86% infants) and not season-dependent."

There is still no information of the quality control. Just stating that the analyses were using standard manufacturers protocols does not ensure robust results, neither does the use of quality control samples. The LC-MS is known to be prone to analytical variability, even when standard protocols are being applied. As the QC samples have been used, it should be straightforward to give the data on the QC analyses. Statement that a batch correction was used, for quite a low number of analyses, raises concerns of the quality of the analyses.

- For the LCMS analysis a commercially available kit was used as well as the accompanying software for data processing, which performed the batch correction (the reviewer may understand this as normalization) steps as part of the standard analytical pipeline. The supplier is committed to quality (<https://biocrates.com/quality-policy/>).

- Batch correction is standard in LCMS-based metabolomic profiling especially large-scale studies. In this analytical run, plasma samples from all four study groups at 12 months and 18 months were analysed (N = 1,558). This would not be considered a low number of analyses in the metabolomics community.
- As requested, the scores plots from the PCA models built on the plasma and urinary metabolites have been added to the supplementary information (Figure S16).
 - The study samples are coloured by batch, highlighting no batch-dependent variation.
 - The QC samples are coloured in black. From the scores plot the reproducibility of the QC samples is clear.
- Collectively, this illustrates the quality, reproducibility and validity of our analysis.

Fig.S16.

A)

B)

Quality control sample and batch distribution of plasma and urinary metabolomes

A) PCA scores plot of plasma metabolites colored by batch number. Median QC per batch colored black. B) PCA scores plot of urinary ¹H NMR spectra colored by batch number. Pooled biological QC per batch colored in black.

Response to reviewers

We are delighted that the reviewers are satisfied with the manuscript. We have made the modest changes requested to the manuscript.

Reviewer #4

The authors present a comprehensive analysis of how birth season shapes the infant metabolome and developmental outcomes in a large Tanzanian cohort. This revision has addressed the major concerns raised previously and substantially strengthened the manuscript in response to Reviewer 1's comments. The distinction between birth-month and sampling-month dependent metabolites is presented with schematics and supplementary figures. Additional analyses and explanations (stratification by sex, inclusion of antibiotic exposure, QC plots, and FDR thresholds) directly address the earlier points.

Reviewer 1 rightly noted that FDR thresholds of 0.05 or 0.01 are more conventional and stringent. The use of an FDR threshold of 0.15 is more lenient, but given the exploratory nature of the study, the consistent seasonal patterns observed for several metabolites (e.g., choline, TMAO, EPA, ceramides) across multiple time points, and the transparent reporting of stricter cut-offs in the supplement, this approach is acceptable.

The manuscript is much stronger, though a few minor refinements would improve clarity:

1. Emphasize that "birth month" functions as a proxy for seasonal exposures rather than a causal factor.
 - A sentence has been included in the discussion to emphasize that birth month is a proxy for the range of season exposures. Other examples of seasonal pressures are listed in the limitations section of the discussion.
 - Line 473-474 "*Birth month represents a proxy for this combination of complex and interacting pressures.*"
2. Make explicit in the main text where stricter FDR thresholds still support the robustness of key findings
 - Throughout the main text, the number of associations that remain following a stricter FDR threshold have been included. The exact p and q values are also included in the Supplementary Tables both including covariates and raw.
 - Line 145 " $n = 3$ for $q < 0.10$ "
 - Line 152 " $n = 5$ for $q < 0.10$ "
 - Line 183 " $n = 4$ for $q < 0.10$ "
 - Line 186 " $n = 4$ for $q < 0.10$ "
 - Line 201 "*for $q < 0.10$, 6M $n = 1$; 12M $n = 11$; 18M $n = 5$* "